EMBO
Molecular Medicine

# MG132-induced progerin clearance is mediated by autophagy activation and splicing regulation

Karim Harhouri[1], Claire Navarro[1], Danielle Depetris[1], Marie-Geneviève Mattei[1], Xavier Nissan[2], Pierre Cau[1,3], Annachiara De Sandre-Giovannoli[1,4] & Nicolas Lévy[1,4,*]

## Abstract

Hutchinson–Gilford progeria syndrome (HGPS) is a lethal premature and accelerated aging disease caused by a *de novo* point mutation in *LMNA* encoding A-type lamins. Progerin, a truncated and toxic prelamin A issued from aberrant splicing, accumulates in HGPS cells' nuclei and is a hallmark of the disease. Small amounts of progerin are also produced during normal aging. We show that progerin is sequestered into abnormally shaped promyelocytic nuclear bodies, identified as novel biomarkers in late passage HGPS cell lines. We found that the proteasome inhibitor MG132 induces progerin degradation through macroautophagy and strongly reduces progerin production through downregulation of SRSF-1 and SRSF-5 accumulation, controlling prelamin A mRNA aberrant splicing. MG132 treatment improves cellular HGPS phenotypes. MG132 injection in skeletal muscle of *Lmna*[G609G/G609G] mice locally reduces SRSF-1 expression and progerin levels. Altogether, we demonstrate progerin reduction based on MG132 dual action and shed light on a promising class of molecules toward a potential therapy for children with HGPS.

**Keywords** autophagy; MG132; PML-NBs; progerin; splicing
**Subject Categories** Genetics, Gene Therapy & Genetic Disease; Neuroscience; Pharmacology & Drug Discovery

## Introduction

Hutchinson–Gilford progeria syndrome (HGPS; OMIM #176670) is a rare genetic disorder, which affects one in 4–8 million children with features of early and accelerated segmental aging. These include growth retardation, thin skin, loss of subcutaneous fat, alopecia, osteoporosis, and myocardial infarction, which is the most frequent cause of death. HGPS patients die at the mean age of 14.6 years (Gordon *et al*, 2014). This accelerated aging disease is caused by a *de novo* synonymous point mutation (c.1824C>T, p.G608G) in exon

11 of the *LMNA* gene encoding A-type lamins (De Sandre-Giovannoli *et al*, 2003; Eriksson *et al*, 2003). This mutation activates a cryptic donor splice site mainly regulated by the serine–arginine-rich splicing factor 1 (SRSF-1) (Lopez-Mejia *et al*, 2011) that causes the in-frame deletion of the last 150 nucleotides of exon 11 in prelamin A pre-mRNAs, leading to the deletion of 50 amino acids at the carboxy-terminal tail of prelamin A, the lamin A precursor. As a result, a truncated, permanently farnesylated prelamin A called progerin accumulates in cell nuclei, where it exerts multiple toxic effects (Goldman *et al*, 2004). At the cellular level, HGPS is characterized by dramatic defects in nuclear envelope structure and function. Primary fibroblasts from HGPS patients exhibit reduced proliferation as well as premature senescence (Goldman *et al*, 2004; Scaffidi & Misteli, 2005, 2008; Vidak & Foisner, 2016). Because lamin A is also a component of the internal nuclear matrix (Vlcek & Foisner, 2007), and because progerin has been shown to heterodimerize with wild-type A- and B-type lamins (Delbarre *et al*, 2006), nucleoplasmic progerin accumulation might affect the distribution and/or the organization of nuclear structures such as nucleoli, speckles, and different nuclear bodies. Moreover, the localization of progerin into intranuclear aggregates in HGPS fibroblast cells was often reported (Goldman *et al*, 2004; Moiseeva *et al*, 2016). Promyelocytic leukemia nuclear bodies (PML-NBs) are discrete nuclear speckles tightly associated with the nuclear matrix (Ascoli & Maul, 1991) and implicated in cellular senescence (Ferbeyre, 2002), whose nuclear arrangement was shown to be influenced by A-type lamin deficiency (Stixova *et al*, 2012).

Unlike the cytosol, where three main complementary proteolytic systems mediate protein degradation, that is, macroautophagy, the ubiquitin proteasome system (UPS), and caspases, in eukaryotic cells' nuclei only the UPS and caspases are known to operate. It has been shown that nucleoplasmic foci mediating protein degradation (Rockel *et al*, 2005) include a subset of the PML-NBs where ubiquitin, proteasomes, and SUMO (small ubiquitin-related modifier) conjugates are concentrated (Lallemand-Breitenbach *et al*, 2008). Along this line, PML-NBs have been proposed to function as sequestration sites of misfolded proteins targeted for proteasomal degradation (Rockel *et al*, 2005). Furthermore, it is intriguing that several proteins involved in cytoplasmic autophagy are physiological

1 Aix Marseille Univ, INSERM, GMGF (Génétique Médicale et Génomique Fonctionnelle), Marseille, France
2 CECS, I-STEM, Institut des cellules Souches pour le Traitement et l'Etude des maladies Monogéniques, AFM, Evry, France
3 AP-HM, Hôpital la Timone, Service de Biologie Cellulaire, Marseille, France
4 AP-HM, Hôpital la Timone, Département de Génétique Médicale, Marseille, France
*Corresponding author. Tel: +33 4 91 32 48 97; Fax: +33 4 91 43 29 90; E-mail: nicolas.levy@univ-amu.fr

 

constituents of PML-NBs, as this is the case for the autophagy-related LC3B protein (He *et al*, 2014), diabetes- and obesity-related protein (DOR) (Mauvezin *et al*, 2012), p62, a cargo receptor for autophagic degradation of ubiquitinated targets (Pankiv *et al*, 2010), and autophagy-linked FYVE protein (ALFY) (Filimonenko *et al*, 2010).

Ubiquitin proteasome system and macroautophagy used to be considered as independent pathways. This view changed recently, in light of findings showing a cross talk between these two major protein degradation systems in a compensatory manner (Wang *et al*, 2013a). Our study thus addressed the question of a potential involvement of PML-NBs in progerin accumulation and/or degradation and of the potential cross talk among the different degradation pathways with respect to progerin accumulation in HGPS fibroblasts.

## Results

### Identification of thread-like PML-NBs, progerin-accumulating compartments, as novel biomarkers in late passage HGPS cell lines

The altered cellular phenotype that characterizes HGPS patients' cells, going along with progerin accumulation, and their premature senescence in culture (Goldman *et al*, 2004; Scaffidi & Misteli, 2005, 2008; Vidak & Foisner, 2016), led us to investigate the subcellular localization of progerin during prolonged fibroblast cell passages.

Immunostaining of progerin in HGPS fibroblasts revealed an extensive accumulation of progerin punctate dots and fibrous aggregates at later passages in culture (Fig EV1A). Given the involvement of PML-NBs in protein sequestration/degradation and the similarity between their structure and that of progerin intranuclear aggregates, we hypothesized that these compartments might be involved in progerin accumulation.

We first explored the localization of PML protein in HGPS fibroblasts and those of age- and passage-matched healthy subjects. At early passages (P9-15), PML protein morphology was similar in patients and controls, with PML-NBs appearing as classical punctate structures (Control: 30 ± 5 vs. HGPS: 24 ± 5) (Fig EV1B). At late passages (passages 16–26), the number of classical PML-NBs decreased (Control: 20 ± 6 vs. HGPS: 8 ± 4) and the number of aberrant ring-like and thread-like PML-NBs appeared and continuously increased in HGPS nuclei, while abnormal PML-NBs were very rarely detected in control fibroblasts, even in more aged cultures (passages 29–33) (Fig EV1B), suggesting that these ring-like and thread-like PML-NBs may be considered as novel disease biomarkers in late passage HGPS cell lines.

We then checked whether progerin colocalized with these structures, and could observe that it was the case (Fig EV1C). Double immunostaining with anti-PML and anti-progerin antibodies on HGPS fibroblasts using confocal microscopy showed that part of progerin intranuclear content colocalized with the PML bodies, seeming to be sequestered inside them in some regions (Fig EV1C, XZ confocal sections).

We next asked whether the physiological protein content of these progerin-associated PML-NBs was maintained. Double immunostaining with antibodies against PML and the main classical components of PML-NBs (SP100, HP1, DAXX, CBP, and ATRX) was performed on HGPS fibroblasts at late passages, showing that these proteins were present, as in wild-type cells (Lallemand-Breitenbach & de The, 2010) (Fig EV1D). The projections of confocal planes and 3D rendering of PML within the nucleus indicate that thread-like PML-NBs are contained inside the nucleoplasm (Fig EV1E). In addition, calreticulin, which localizes into the lumen of the endoplasmic reticulum or the nuclear envelope, has never been observed within PML-NBs neither in HGPS fibroblasts nor in control cell lines (Fig EV1E, lower panel). This suggests that PML-NBs do not correspond to an invagination of the nuclear envelope.

To determine whether other nuclear lamina-resident proteins, in addition to progerin, had the capacity to be targeted to the PML-NBs, we tested the subcellular localization of lamins A, C, B1, B2, Nup-153, and emerin. Interestingly, lamin A, lamin C, and Nup-153 were excluded from the PML-NBs while, among the proteins explored, B-type lamins (B1 and B2) and emerin were also included within the ring-like and thread-like progerin containing PML-NBs structures (Fig EV2A and B).

### MG132 decreases progerin levels in HGPS cells

The presence of ubiquitin and 26S proteasome into the thread-like PML structures (Fig EV2C) strongly supports that they might be involved in the degradation of proteins and in particular progerin via the UPS.

To determine whether proteasome inhibitors may influence the levels of progerin, we performed a dose–response analysis of MG132 treatment on HGPS fibroblasts for 24 h and investigated progerin levels by immunofluorescence. Unexpectedly, instead of increasing progerin levels, the treatment resulted in a decrease of progerin staining intensity starting from 5 μM MG132 treatment (Fig 1A). Since viability tests showed that this dose did not induce significant cell mortality in treated HGPS fibroblasts (Fig 1B), while cell viability showed partial reduction at 10 μM, we carried out Western blot experiments to investigate the kinetics of 5 μM MG132 treatment on progerin levels in HGPS fibroblasts. As shown in Fig 1C, when normalized with a tubulin control, protein quantification revealed a ~31, 58, and 65% reduction in progerin amounts at 6, 24, and 48 h, respectively, in MG132-treated HGPS cells as compared to DMSO-treated HGPS cells. To exclude that upon MG132 addition, progerin may become insoluble and be incompletely extracted from aggregates, and cell proteins lysates were solubilized with urea. To confirm or rule out the involvement of proteasome inhibition in progerin clearance, we tested several proteasome inhibitors in the same conditions. The boronate analogue MG262 (Z-Leu-Leu-Leu-boronate) was less effective (Fig EV3A and C). MG115 (Z-Leu-Leu-nVal-al), another aldehyde proteasome inhibitor, initially thought as being less potent than MG132, was found as having a similar efficacy in suppressing progerin accumulation (Fig EV3C). We next tested the effect of bortezomib and carfilzomib, two FDA-approved proteasome inhibitors (treatment of multiple myeloma) (Herndon *et al*, 2013), on progerin clearance. Unlike the aldehyde MG132 and the boronate MG262, these two drugs did not induce an effective progerin clearance since progerin expression was maintained upon commonly used concentrations (Fig EV3B) or slightly reduced after exposure to high concentrations (Fig EV3C). It is known that these two drugs

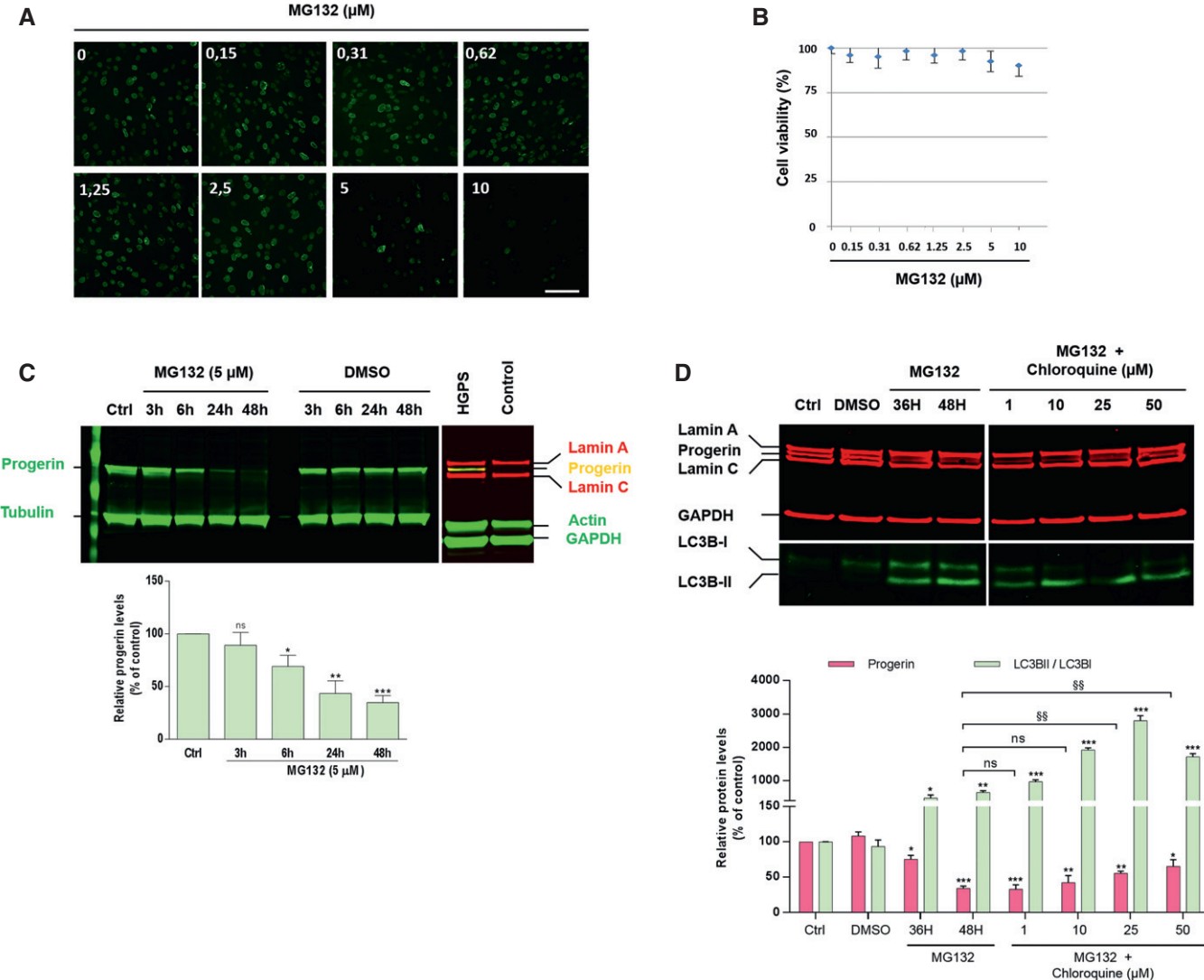

**Figure 1.  MG132 reduces progerin levels in HGPS fibroblasts.**

A   Immunofluorescence images of HGPS fibroblasts treated with MG132 for 24 h and stained for progerin (green). Scale bar, 40 μm. ($n$ = 4).

B   HGPS fibroblasts viability was measured at 24 h post-treatment with MG132 at the indicated concentrations using CellTiter-Glo Luminescent Cell Viability Assay. Results are reported as viability percentage of MG132-treated cells relative to DMSO-treated cells. ($n$ = 4).

C   MG132 treatment results in a decrease of progerin levels, (Ctrl: untreated HGPS cells). Lower panel, progerin expression levels in MG132-treated cells were normalized to those treated with DMSO and to tubulin values (loading control). Fibroblasts isolated from biopsies of healthy subjects (Control) were used as negative controls to validate the specificity of anti-progerin antibody. ($n$ = 5).

D   Upper panels, lamin A/C, progerin, GAPDH, LC3B-I and LC3B-II expression, in whole lysates from HGPS fibroblasts untreated (Ctrl) or treated with DMSO (48 h), 5 μM MG132 (36 and 48 h), or both 5 μM MG132 and chloroquine added simultaneously (48 h). The images were cropped from the same Western blot experiment. Lower panels, progerin expression levels (relative to DMSO-treated cells: "*", or relative to MG132-treated cells for 48H: "§", and LC3BII/LC3BI ratios relative to DMSO-treated cells: "*") were normalized to GAPDH values. ($n$ = 5).

Data information: Results are expressed as mean ± SEM, Student's *t*-test, *$P$ < 0.05, **$P$ < 0.01, §§$P$ < 0.01, ***$P$ < 0.001, experimental vs. control; the exact $P$-values are indicated in Appendix Table S1.

Source data are available online for this figure.

(bortezomib IC50: 0.6 nM and carfilzomib IC50: 5 nM) inhibit proteasome activities more effectively than MG132 (IC50: 100 nM) (Kisselev & Goldberg, 2001); using a proteasome activity evaluation kit, we could confirm that these drugs maintained a similar ability to inhibit proteasome activity in HGPS cells (Fig EV3D).

Promyelocytic leukemia protein is degraded through a SUMO-triggered RNF4/ubiquitin-mediated pathway and most

PML-associated proteins undergo SUMO-conjugation (Lallemand-Breitenbach *et al*, 2008). On the other hand, some reports have demonstrated sumoylation of lamin A at lysine 201 (Zhang & Sarge, 2008). This sumoylation site is also present in progerin and could be involved in its translocation to PML-NBs. As shown in Appendix Fig S1A, MG132-treated HGPS cell extracts were immunoprecipitated with an anti-progerin antibody and then immunoblotted with an

anti-lamin A/C antibody; this experiment detects progerin and its partner proteins (lamin A and lamin C) under non-denaturing conditions (Appendix Fig S1A, left) and only progerin, not lamin A/C, in denaturing conditions (Appendix Fig S1A, right). Under non-denaturing conditions, we detected high molecular weight bands indicative of conjugated lamin A, lamin C, and progerin. Western blot using SUMO 2/3 antibody revealed that these bands corresponded to their sumoylated isoforms (Appendix Fig S1B), corroborating sumoylated progerin localization within PML-NBs.

Using the same co-immunoprecipitation techniques (Appendix Fig S1C), we next determined whether progerin is ubiquitinated. Under non-denaturing conditions, we observed that sumoylated high molecular weight bands also corresponded to ubiquitinated lamin A, lamin C, and progerin isoforms. However, under denaturing conditions, only a specific band of lower molecular weight was present, corresponding to ubiquitinated progerin. This experiment showed that MG132 elicited progerin decrease only when this was ubiquitinated, but not when it was both sumoylated and ubiquitinated. This effect seems specific for ubiquitinated progerin, since MG132 treatment globally induces an increase of ubiquitinated proteins both in control and HGPS cells (Appendix Fig S1D). To better determine whether progerin sumoylation is required for its clearance, we treated HGPS fibroblasts with sumoylation inhibitors (either 2D08 or ginkgolic acid) in combination with proteasome inhibitors (MG132, MG262, MG115, bortezomib, or carfilzomib). The results presented in Appendix Fig S1E show that progerin levels are not significantly affected in the presence or absence of sumoylation inhibitors in response to treatment with MG132, MG115, or bortezomib. Under the same conditions, the effect of MG262 on progerin clearance is more pronounced when sumoylation is inhibited. On the contrary, sumoylation of progerin appears to be required to achieve its clearance in response to carfilzomib.

**Progerin clearance is partially due to autophagy**

The observed progerin decrease upon proteasome inhibition suggested that progerin could be a substrate for autophagic degradation, since it is known that proteasome inhibitors induce autophagy as a compensatory response (Zhu *et al*, 2010; Zang *et al*, 2012; Tang *et al*, 2014). In order to assess whether macroautophagy was involved in MG132-enhanced progerin clearance, we used chloroquine as an autophagy inhibitor. Chloroquine alters the acidic pH of lysosomes and leads to inhibition of both fusion of autophagosomes with lysosomes and lysosomal protein degradation (Shintani & Klionsky, 2004). Treatment of MG132-exposed HGPS fibroblasts with high-dose chloroquine partially inhibited the effect of MG132 and induced a partial increase of progerin levels (Fig 1D), suggesting that the increased clearance of progerin observed with MG132 treatment was mediated at least in part by the autophagic-lysosomal pathway.

LC3B was used to evaluate the macroautophagy level since the conversion of LC3B-I to LC3B-II by lipidation provides an indicator of autophagic activity (Ichimura *et al*, 2000). As expected, LC3B-II/LC3B-I ratios are increased following MG132 treatment, concomitantly with the decrease of progerin levels (Fig 1D).

To investigate subcellular progerin localization upon MG132 treatment and its possible dynamics, we carried out time-course indirect immunofluorescence experiments using progerin and PML antibodies on HGPS cell lines treated with MG132 (5 µM) for 6 and

24 h. As shown in Fig 2A, 24-h treatment with MG132 induces the formation of large progerin and PML intranuclear foci. In order to investigate whether these foci colocalize with the nucleolus, we carried out a similar experiment, using PML, emerin, p53, lamin B, the 26S proteasome subunit, and ubiquitin antibodies on HGPS and control cell lines treated with MG132 (5 µM) for 24 h, together with fibrillarin antibodies staining the dense fibrillar component of the nucleolus. In both fibroblasts from healthy subjects and HGPS patients, fibrillarin staining showed an overlap with PML as well as with all other tested proteins (Fig 2B), including the 26S proteasome subunit and ubiquitin.

To further characterize the involvement of autophagy in progerin dynamics and clearance, we performed co-immunostaining of progerin, PML, and LC3B upon MG132 treatment. We detected for PML and LC3B both a nucleolar and a cytosolic colocalization upon 24 h MG132 treatment (Fig 3A). While, during this treatment period, progerin mainly remained localized within the nucleolus (Fig 3B), after 48 h, progerin aggregates were detected into the cytosol where we could observe its colocalization with the autophagic marker LC3B, p62, a known autophagy substrate, and LAMP-2, a lysosomal membrane protein (Fig 3C). Taken together, these results suggest that, upon MG132 treatment, progerin accumulated into autophagic vacuoles and thus could undergo autophagic degradation in the cytoplasm.

**Involvement of other pathways in the reduction of progerin levels upon MG132 treatment**

Upon MG132 treatment, the incomplete restoration of progerin levels after blockade of autophagy using either chloroquine and bafilomycin A, an inhibitor of lysosomal proton pump that prevents maturation of autophagic vacuoles by inhibiting fusion between autophagosomes and lysosomes (Yamamoto *et al*, 1998) (Fig 4A), suggests that MG132 efficiency on progerin clearance may be mediated, in addition to autophagy, by another concomitant activity.

Caspase-6 is widely recognized as being responsible of lamin A/C cleavage during apoptosis. Lamin A/C as well as progerin exhibits a conserved VEID caspase-6 cleavage site (Slee *et al*, 2001). To investigate the involvement of caspase-6 in MG132-induced progerin clearance, we treated HGPS cells for 48 h with a combination of MG132 and a caspase-6 inhibitor. Again, progerin levels remained below the HGPS control even in the presence of autophagy inhibition by chloroquine (Fig 4A), suggesting that another pathway, in addition to autophagy, was involved in progerin levels' reduction upon MG132.

In order to continue investigating the pathways involved in progerin clearance and since progerin has a leucine-rich nuclear export signal (NES) involved in the exportin-1 dependent nuclear export through the nuclear pores (Fukuda *et al*, 1997), we inhibited this process using leptomycin B and showed that MG132-induced progerin clearance is not dependent on its cytoplasmic export through the nuclear pores (Fig 4A).

**MG132 reduces prelamin A aberrant mRNA splicing by decreasing SRSF-1 and increasing SRSF-5 levels**

Since the blockade of all main degradation pathways was not able to restore basal progerin level as compared to HGPS control cells,

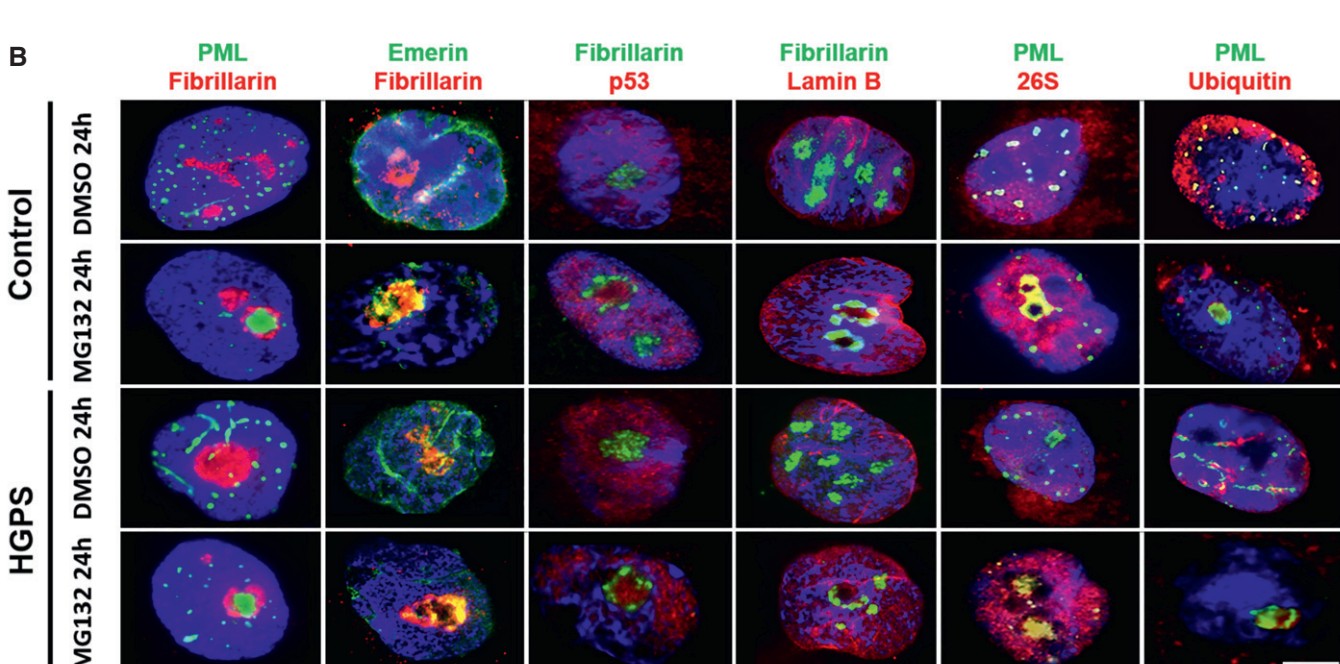

**Figure 2.  Nucleolar translocation of progerin upon MG132 treatment.**

A   Subnuclear distribution of progerin (red) and PML (green) after MG132 treatment of HGPS fibroblasts. Cells cultured with 5 µM MG132 for 24 h show staining of progerin and PML in intranuclear foci.

B   MG132 induces the translocation of PML (green), emerin (green), p53 (red), 26S proteasome subunit (red), and ubiquitin (red) into nucleoli of HGPS and control fibroblasts cultured in the presence of 5 µM MG132 for 24 h (relative to DMSO-treated cells: control). Nucleoli were labeled with fibrillarin antibodies and nuclei with DAPI (blue). The merged images are shown.

Data information: Data in (A,B) are representative of six independent experiments. The experiments were performed on fibroblasts of HGPS patients and healthy subjects matched for age and passage. Scale bars, 5 µm.

we further evaluated whether MG132 could have an effect at the transcriptional level, and performed quantitative reverse transcription–polymerase chain reaction (qRT–PCR) assays to quantify progerin transcript levels in MG132-treated and control HGPS cells. As shown in Fig 4B, MG132 treatment almost extinguished progerin mRNA expression in a time dependent manner (the strongest activities were observed at 24 and 48 h after treatment), confirming the hypothesis of MG132 effect on progerin transcription, in addition to its effect at the protein level.

It has been previously shown that the serine–arginine-rich (SR) splicing factors are modulators of the utilization of the *LMNA* and progerin 5′ splice site (5′SS), favoring the production of progerin instead of lamin A from c.1824C>T-mutated *LMNA* alleles (SRSF-1) (Lopez-Mejia *et al*, 2011), enhancing the use of the *LMNA* 5′SS at the expenses of progerin 5′SS (SRSF-6 and SRSF-5) (Lopez-Mejia

*et al*, 2011; Vautrot *et al*, 2016), or lowering both lamin A and progerin production (SRSF-2) (Lee *et al*, 2016). We thus hypothesized that MG132 effect could be mediated by SR proteins modulation and, therefore, a reduced use of the aberrant splicing site leading to progerin mRNA production. Interestingly, we show that the treatment of HGPS fibroblasts with MG132 resulted in a significant decrease of SRSF-1 levels concomitantly with the decrease of progerin levels at 24 and 48 h post-treatment. At 72 and 96 h post-treatment, we observed a normalization of progerin and SRSF-1 levels (Fig 4C). Interestingly, the loss of MG132 activity in inhibiting the proteasome parallels its inefficiency in reducing progerin and SRSF-1 levels at these prolonged post-treatment time points (Fig 4D). By renewing treatment after 48 h, progerin and SRSF-1 levels remain down and SRSF-5 levels increase during 4–12 h post-treatment, after which, they normalize (Fig 4E). Using a pan-specific

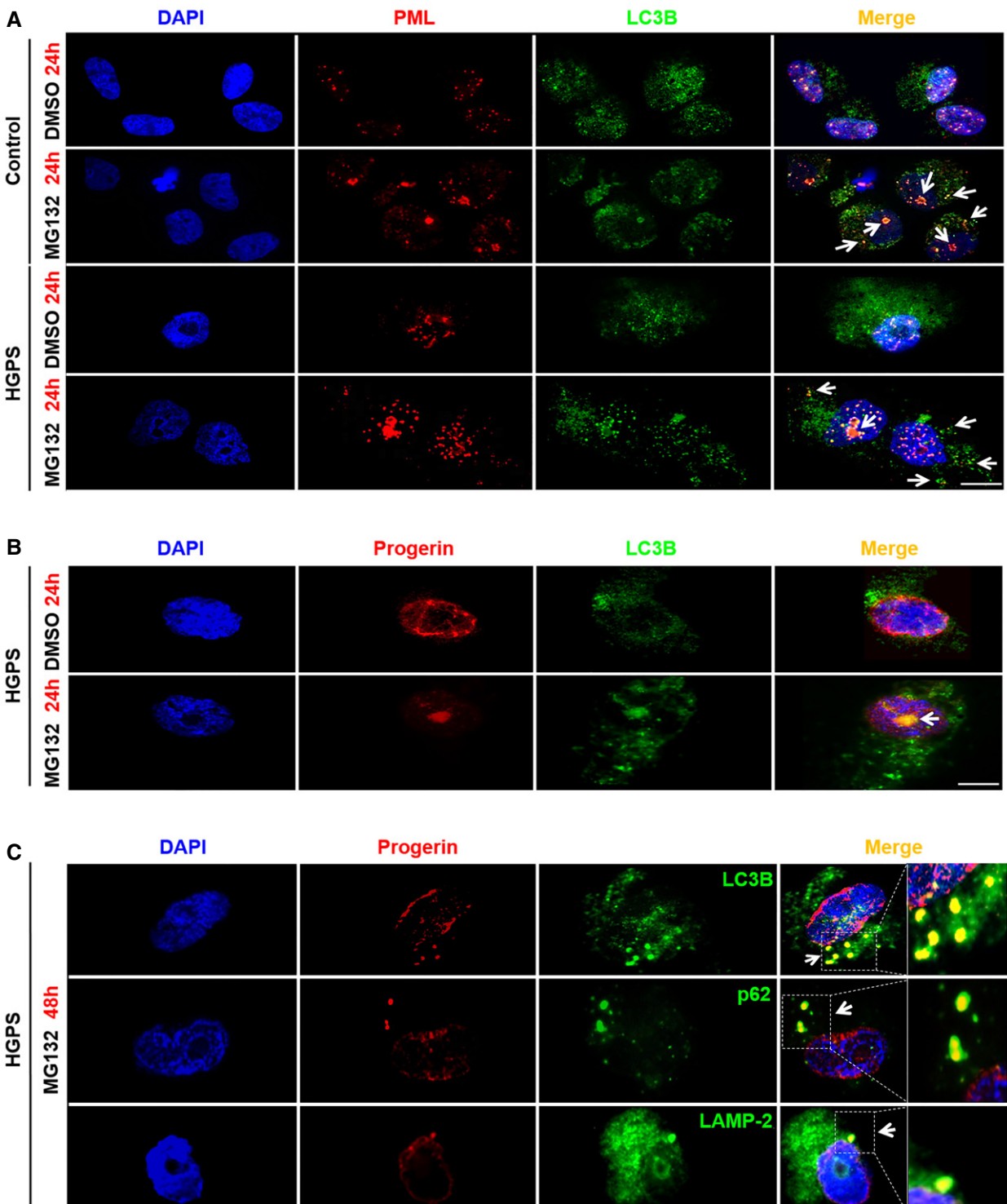

**Figure 3. Progerin accumulates in autophagic vacuoles upon MG132 treatment.**

A   Immunofluorescence staining of PML (red) and LC3B (green) in HGPS and control fibroblasts both treated with MG132 (5 μM) or DMSO (control) for 24 h. Nucleolar and cytoplasmic colocalization of LC3B with PML are shown (arrows). Scale bar, 10 μm.

B   Nucleolar progerin (red) and LC3B (green) accumulation (arrows) in HGPS fibroblasts upon 5 μM MG132 treatment. Scale bar, 5 μm.

C   Progerin (red) accumulated in LC3B (green), p62 (green) and LAMP-2 (green) positive cytoplasmic vesicles (arrows) after 48 h MG132 (5 μM) exposure of HGPS fibroblasts. Scale bar, 5 μm. The magnifications are shown; scale bar, 200 nm.

Data information: Data in (A–C) are representative of five independent experiments. The experiments were performed on fibroblasts of HGPS patients and healthy subjects matched for age and passage.

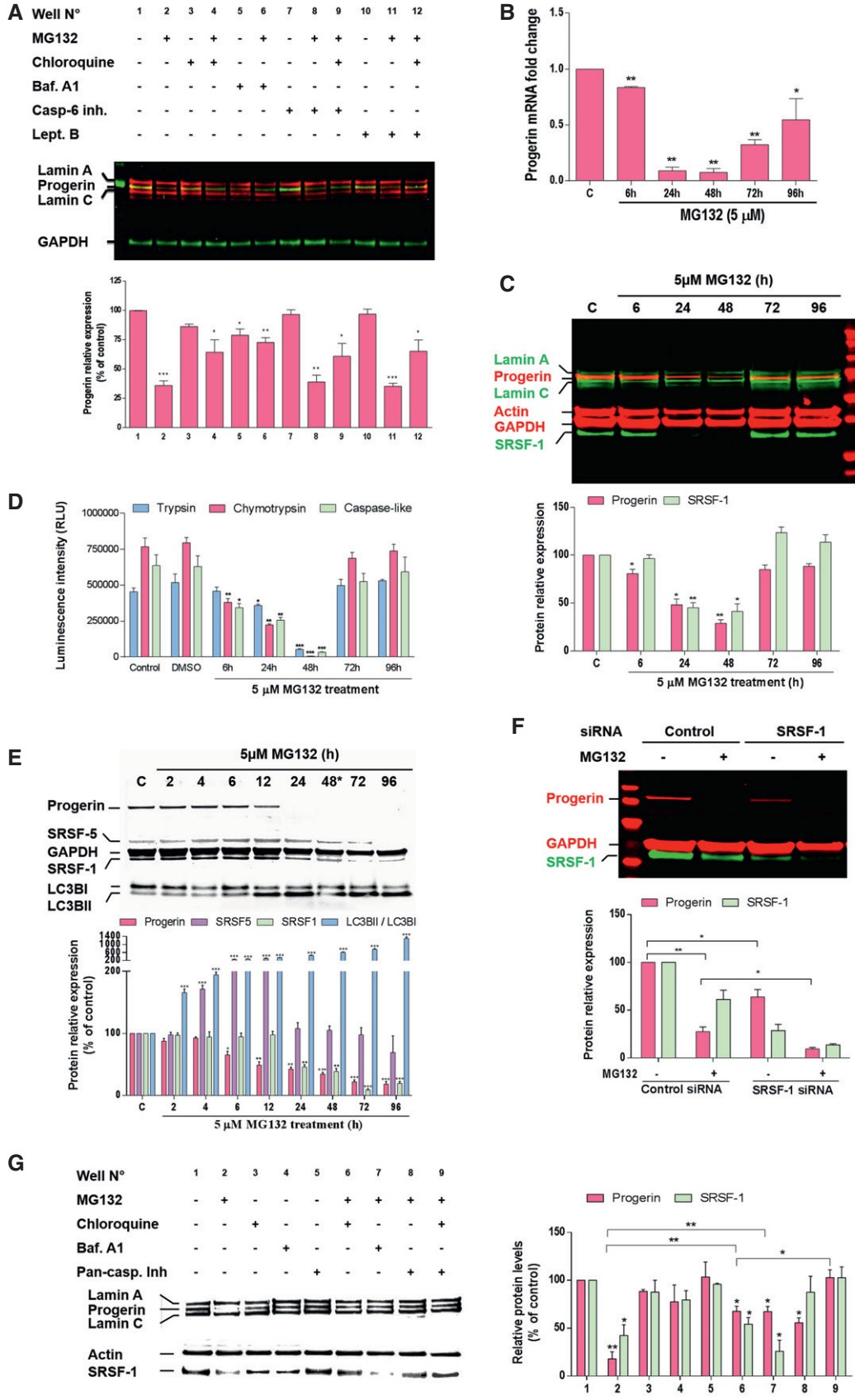

Figure 4.

◄

**Figure 4.  MG132 reduces progerin transcripts and splicing factor SRSF-1 protein expression, while SRSF-5 protein levels are increased.**

A   Upper panel, lamin A/C, progerin, and GAPDH expressions in whole lysates from HGPS fibroblasts treated for 48 h either with DMSO control (−) or with (+) the indicated drug (s) (MG132 (5 μM), chloroquine (50 μM), bafilomycin A1 (Baf. A1) (100 nM), caspase-6 inhibitor (Casp-6 inh.) (50 μM) or leptomycin B (Lept. B) (20 ng/ml). Lower panels, progerin expression levels in drug (s)-treated cells relative to DMSO-treated cells (well no. 1) were normalized to GAPDH values. (*n* = 5).

B   Downregulation of progerin transcripts in response to MG132. Quantitative real-time PCR analyses of progerin mRNA levels in HGPS fibroblasts treated with 5 μM MG132 relative to DMSO-treated cells (Control: "C"). (*n* = 3).

C   A representative Western blotting experiment in whole lysates of HGPS fibroblasts showing progerin, lamin A, lamin C, actin, GAPDH and SRSF-1 expression in MG132-treated HGPS cells up to 96 h and relative to DMSO-treated cells "C". Urea was used to lyse cells. (*n* = 4).

D   Proteasome activities in HGPS fibroblasts treated with 5 μM MG132 relative to DMSO-treated cells, (Control: untreated cells). (*n* = 3).

E   MG132-mediated SRSF-5 accumulation and SRSF-1 downregulation correlate with progerin clearance. A representative Western blotting experiment in whole lysates of HGPS fibroblasts showing progerin, SRSF-5, GAPDH, SRSF-1, LC3BI, and LC3BII expression in MG132-treated HGPS cells up to 96 h and relative to DMSO-treated cells (Control: "C"). The medium was replaced with new drug every 48 h. Urea was used to lyse cells. (*n* = 4).

F   siRNA inactivation of SRSF-1 reduces progerin levels in HGPS fibroblasts. HGPS fibroblasts were transfected with control siRNA or with siRNA specific for SRSF-1 and 48 h later cells were treated for 24 h with DMSO control (−) or with (+) MG132 (5 μM). (*n* = 3).

G   Left panels, caspase-mediated downregulation of SRSF-1, in addition to autophagy, contribute to progerin clearance. Western blotting evaluation of lamin A/C, progerin, actin and SRSF-1 in whole lysates from HGPS fibroblasts treated for 48 h either with DMSO control (−) or with (+) the indicated drug (s) [MG132 (5 μM), chloroquine (50 μM), bafilomycin A1 (100 nM) or pan-caspase inhibitor (50 μM)]. Rihgt panels, progerin and SRSF-1 expression levels relative to DMSO-treated cells (well no. 1) were normalized to actin values using ImageJ software. (*n* = 6).

Data information: Results are expressed as mean ± SEM, Student's *t*-test, *\*P* < 0.05, *\*\*P* < 0.01, *\*\*\*P* < 0.001, experimental vs. control; the exact *P*-values are indicated in Appendix Table S1.

Source data are available online for this figure.

antibody to SR proteins, we confirmed SRSF-1 and SRSF-5 modulation by MG132 treatment while the levels of other SR proteins are not altered (data not shown).

To validate the involvement of SRSF-1 in regulating the expression of progerin, we treated HGPS cells with small-interfering RNAs (siRNAs) to transiently deplete SRSF-1 cell content. The results presented in Fig 4F show that siRNA knockdown of SRSF-1 greatly reduces progerin levels, evidencing a major role of this splicing factor in MG132-induced progerin levels reduction.

Since SRSF-1 and SRSF-5 are involved in the control of alternative lamin A/progerin splicing, we hypothesized that MG132 treatment modulated lamin A transcript expression. Analysis of qRT–PCR upon MG132 time-course treatment revealed an expected significant decrease of lamin A transcript with a much less pronounced decrease of lamin C transcripts (Fig EV4A). As shown in Fig EV4B, we observed that both lamin A and C protein levels were stable or increased during 48 h treatment with MG132.

Toward exploring whether mechanisms involving caspases play a role in SRSF-1 reduction upon MG132 treatment, we added the pan-caspase inhibitor and observed a rescue of SRSF-1 protein expression that almost reached the control levels, concomitantly with an increase of progerin levels (Fig 4G, well no. 8). Finally, the combined use of the pan-caspase inhibitor and the autophagy inhibitor chloroquine upon MG132 treatment allowed rescuing basal levels of progerin re-expression (Fig 4G, well no. 9).

Taken together, our results indicated that autophagy induction and splicing mechanisms through SRSF-1 depletion and a rapid but not maintained SRSF-5 accumulation played converging roles toward progerin clearance under MG132 treatment.

## MG132 treatment ameliorates cellular HGPS phenotypes

One of the hallmarks of primary fibroblasts from HGPS patients is the abnormal nuclear shapes due to nuclear envelope blebbing (Goldman *et al*, 2004). Using lamin A/C and DAPI staining, these characteristic nuclear defects have been explored to determine the effectiveness of MG132 treatment toward reversing HGPS nuclei abnormalities (Fig 5A). In comparison with the passage-matched

and DMSO-treated HGPS cells, MG132-treated HGPS cells exhibited a significant reduction in nuclear blebbing (Fig 5B).

Cellular senescence is also considered as a major hallmark of HGPS, as well as of normal aging cells. We first assessed senescence in HGPS and control fibroblasts at the same culture passage. As shown in Fig 5C, HGPS cells exhibit an increased senescence phenotype, measured by β-galactosidase quantification, compared to cells of an age-matched control, but less pronounced than that of fibroblasts from an advanced-age healthy subject. Interestingly, all HGPS and control cells treated with MG132 for 48 h exhibited a decreased senescence rate (Fig 5C). We then performed MG132 treatment up to 10 days every 48 h on HGPS cell lines and found a statistically significant decrease in the number of senescent cells when compared to DMSO-treated cells (Fig 5D). Furthermore, we observed that this MG132 treatment scheme increased the number of proliferating cells relative to DMSO-treated cells with a curve approximately similar to that of normal cells (Fig 5E), as well as viable cells (Fig 5F), while inducing either no or slight decrease of cytotoxicity on HGPS cells (Fig 5G). Other characteristics of fibroblasts from individuals with HGPS result in an increase in the number of γ H2AX (phosphorylated histone H2AX, a marker of DNA double-strand breaks) and loss of the heterochromatin marker tri-methyl lysine 9 of core histone H3 (Tri-Me-K9) (Zhang *et al*, 2016) as well as a reduction in lamina-associated polypeptide (LAP2α), lamin B1 or the heterochromatin adaptor between the nuclear lamina and chromatin (HP1α) (Scaffidi & Misteli, 2005). Importantly, treatment of HGPS cells with MG132 improved significantly these cellular phenotypes (Fig 5H). To further investigate the restoration of normal cellular function of HGPS fibroblasts upon MG132 treatment, we performed RNA-seq experiment (accession number: E-MTAB-5807) and analyzed the expression levels of genes that are misregulated in HGPS fibroblasts. Interestingly, we found an increase in the transcripts levels of genes involved in autophagy activation (p62, ATG4A, ATG4B, ATG4D, ATG3, ATG14, LLK1), proteasome subunits production (PSMC4, PSMB3), splicing (SRSF5), metalloprotease transcripts including MMP3 (whose levels are decreased in HGPS cells), growth factors (EGF, FGF17, FGF22), apoptosis inhibitors (BAG3, BFAR) as well as the improvement of other transcripts involved in inflammation (IkB, SIRT6) (Fig 5I).

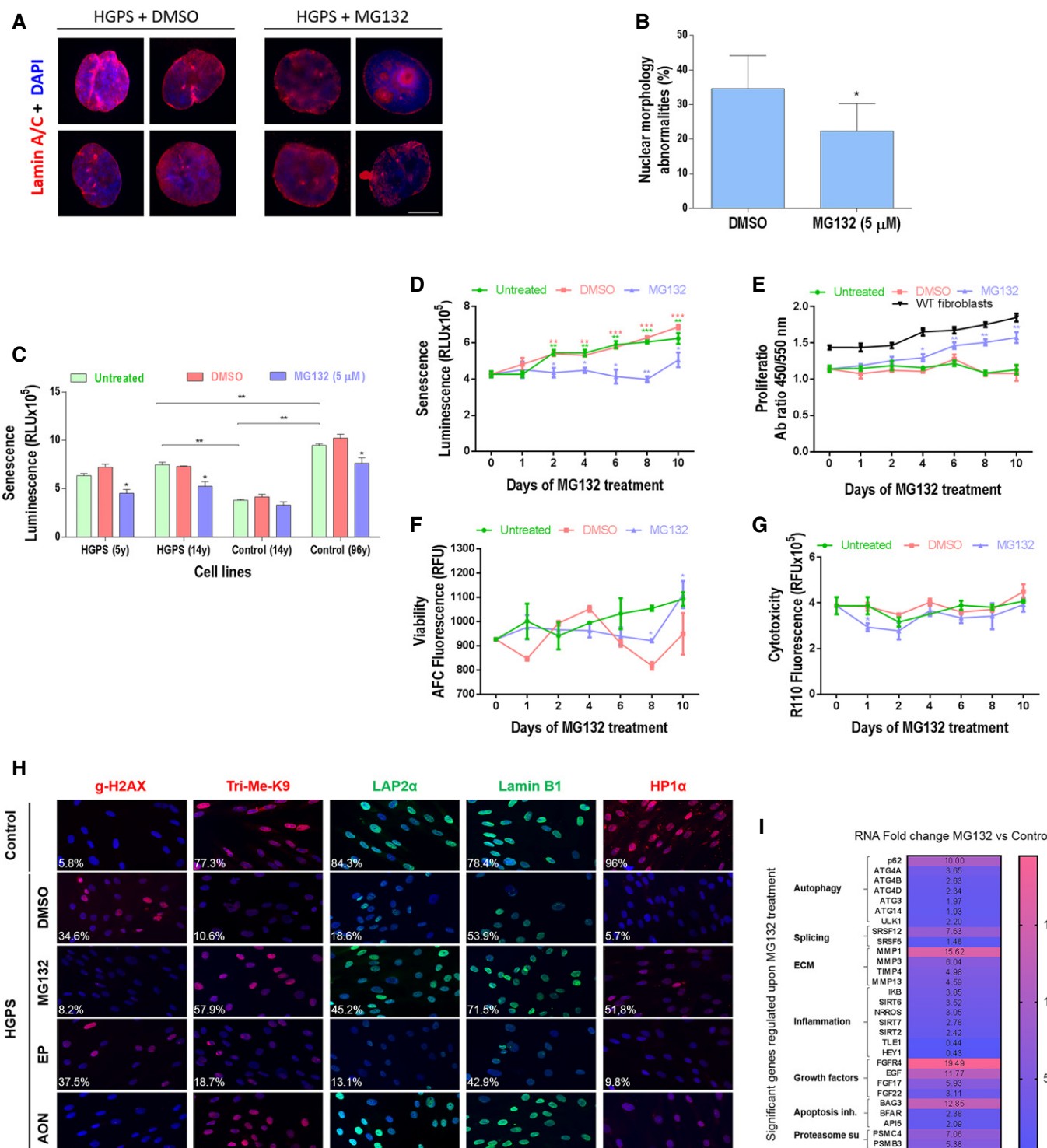

Figure 5.

**Progerin clearance upon MG132 treatment is observed in HGPS iPSC-derived cell lines as well as *in vivo* in the *Lmna*^G609G/G609G mouse model**

To determine whether MG132 treatment also influenced the level of progerin in other cell lineages, we used previously generated iPSC from HGPS patients' fibroblasts (Nissan *et al*, 2012). Mesenchymal stem cells (MSC) and vascular smooth muscle cells (VSMC) derived from HGPS iPSC presented the commonly described abnormalities of HGPS fibroblasts, namely loss of proliferation capacities, premature senescence, and nuclear blebbing (Zhang *et al*, 2011; Nissan *et al*, 2012). As shown in Fig 6A, MG132-treated VSMC showed a

**Figure 5.  MG132 improves cellular HGPS phenotypes.**

A    Immunofluorescence images of HGPS fibroblasts cultured with DMEM medium containing 5 μM MG132, or the same volume of vehicle (DMSO, 0.025% v/v) for 48 h and stained for lamin A/C (red) and DAPI (blue). Scale bar, 5 μm. ($n = 3$).

B    The percentage of normal nuclei (nuclei with a smooth oval shape) and abnormal nuclei (nuclei with blebs, irregular shape, or multiple folds) was calculated using the Nuclear Irregularity Index (NII) plugin of the ImageJ software (version 1.6.0, NIH, USA). At least 200 fibroblast nuclei were randomly selected for each cell line. A representative image and the mean values of three different experiments are shown.

C    Senescence rate in 2 HGPS fibroblasts and two control fibroblasts treated with 5 μM MG132 for 48 h relative to DMSO-treated cells. Each experiment was performed on cells at the same passage level. Senescence is determined as relative light units (RLU). ($n = 4$).

D–G    HGPS fibroblasts were untreated, vehicle control (DMSO) or 5 μM MG132 treated up to 10 days. The medium was replaced with new drug every 48 h. ($n = 3$). Senescence rate in HGPS fibroblasts (untreated or DMSO-treated cells at each time point vs. Day 0. MG132-treated cells vs. DMSO-treated cells at each time point) (D). Cell proliferation rate based on the incorporation of bromodeoxyuridine (BrdU) into the DNA was expressed as absorbance OD450-550. (MG132-treated cells vs. DMSO-treated cells at each time point). Proliferation rates in normal fibroblasts (WT fibroblasts) are presented (E). Viability (MG132-treated cells vs. DMSO-treated cells at each time point) (F) and cytotoxicity (MG132-treated cells vs. DMSO-treated cells at each time point) (G).

H    Immunofluorescence microscopy on primary dermal fibroblasts from a healthy individual (control) and an individual with HGPS treated with DMSO (MG132 vehicle control), 5 μM MG132, 20 μM control scrambled morpholino antisense oligonucleotides (Scr-AON), or 20 μM specific AON for 10 days. The medium was replaced with new drug every 48 h. Cells were stained with DAPI (blue) and antibodies to the indicated proteins. The percentage of positive staining is indicated, at least 200 fibroblast nuclei were randomly selected for each cell line ($n = 3$) and examined using ImageJ software. Scale bar, 40 μm.

I    Heatmap of RNAseq data (ArrayExpress accession number: E-MTAB-5807) from HGPS fibroblasts treated with DMSO (vehicle control) or with 5 μM MG132 for 6 h. This analysis represents the fold change, in MG132-treated relative to DMSO-treated HGPS fibroblasts, of the most characteristic transcripts of the cellular HGPS phenotype. ($n = 2$).

Data information: Results are expressed as mean ± SEM, Student's *t*-test, \*$P < 0.05$, \*\*$P < 0.01$, \*\*\*$P < 0.001$, experimental vs. control; the exact *P*-values are indicated in Appendix Table S1.

significant reduction of progerin staining starting at 1.25 μM. In parallel, we compared progerin and SRSF-1 levels upon MG132 treatment among HGPS fibroblasts, iPS-MSC, and iPS-VSMC. In all the tested cell lines, both progerin and SRSF-1 levels were decreased by MG132 treatment (Fig 6B). As well, in all the cell lines, we observed an increased LC3B-I to LC3B-II autophagic switch upon MG132 treatment. Furthermore, viability tests were performed on both HGPS iPS-MSC and iPS-VSMC cell lines, in order to compare their sensibility to MG132 treatment to that of HGPS fibroblasts, showing that iPS-MSC tolerated similarly the treatment, while iPS-VSMC were less resistant to high MG132 doses (Fig 6C). Additionally, these results indicated that progerin clearance observed in HGPS iPS-MSC and iPS-VSMC cell lines was not linked to increased apoptosis or cell death, since the doses used in this study did not induce significant cell mortality (Fig 6C).

Because MG132 treatment induced an almost complete clearance of progerin in various cell lines, it was of interest to investigate its effect *in vivo*. We thus performed a pilot study using MG132 (at 0.04, 4, 7, 15, and 30 mg/kg), which was injected daily for 2 months in *Lmna^{G609G/G609G}* mice either intravenously or intraperitoneally: Progerin expression levels were then compared in treated mice ($n = 5$ for each dose) vs. placebo mice ($n = 5$). Analysis of progerin levels in different tissues, including heart, liver, spleen, kidney, lung, and skeletal muscle before and after treatment, showed that baseline progerin levels as well as progerin reduction upon treatment varied from one mouse to another and even in different tissues of the same mouse. Therefore, the reduction of progerin levels upon systemic treatment was not significant.

Given these preliminary results and wishing to evaluate MG132 effects on progerin levels independently from confounding systemic metabolic events, we chose a local approach. To this end, we performed intramuscular injections in our knock-in progeria mouse model (*Lmna^{G609G/G609G}*) carrying the c.1827C>T (p.Gly609Gly) mutation (Osorio *et al*, 2011). The drug was injected three times per week for 2 weeks in the right gastrocnemius of the two MG132-treated groups (1 μg/kg, $n = 5$ and 10 μg/kg, $n = 5$). The left gastrocnemius of each mouse was used as its own control: this

approach freed our observation from the variable baseline progerin levels observed among mice and among different organs. As shown in Fig 7, treatment with MG132 induced a significant decrease of progerin levels, concomitantly with a significant decrease of SRSF-1 levels, in the treated muscle compared to the untreated contralateral muscle at both treatment doses. Upon treatment, the levels of lamin A remained constant or slightly increased upon both doses, while lamin C levels were significantly decreased upon treatment with the higher MG132 dose. It should also be mentioned that, when mice were injected with MG132, no deleterious effects were observed (no necrosis or inflammation, nor tumor formation at the injection sites and no weight loss or premature death upon local treatment).

## Discussion

It has been largely proven that HGPS pathophysiology mainly relies on the intranuclear ubiquitous accumulation of progerin, a toxic lamin A truncated derivative. The present study focused on the dynamics of progerin localization and the putative involvement of the three main protein degradation systems (proteasome, macroautophagy, and caspases), in its clearance, which is overtly deficient in HGPS cells. These cell lines exhibit progressive nuclear abnormalities that parallel progerin accumulation with increasing cell passages, and, for the first time, we showed that progerin also accumulates into atypical nucleoplasmic ring-like and tubular aggregates. We hypothesized that PML-NBs might be involved in progerin accumulation, since they contain proteasome and ubiquitinated proteins, being implicated in protein sequestration/degradation processes. Indeed, our data provide evidence that progerin is sequestered into PML-NBs within ring-like and thread-like structures that may be considered as novel biomarkers in cultured late passage HGPS cell lines. Structural changes within PML-NBs have been shown to occur during DNA and RNA viral infections [reviewed in (Doucas & Evans, 1996; Sternsdorf *et al*, 1997)] since PML-NBs are sites for the early stages of transcription and replication of DNA and RNA viruses and are also sites for the subsequent

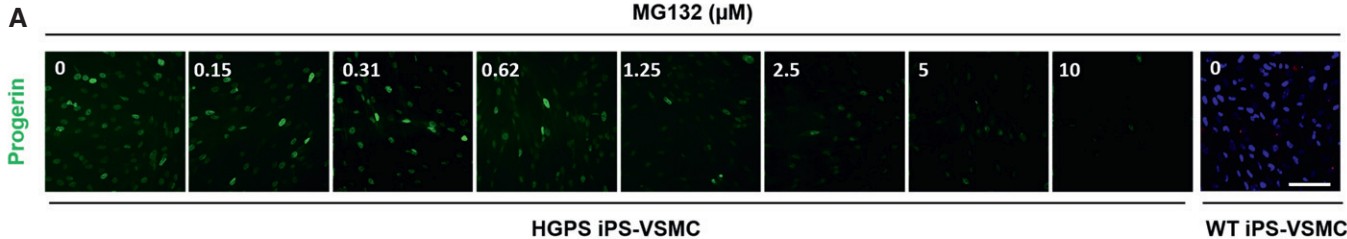

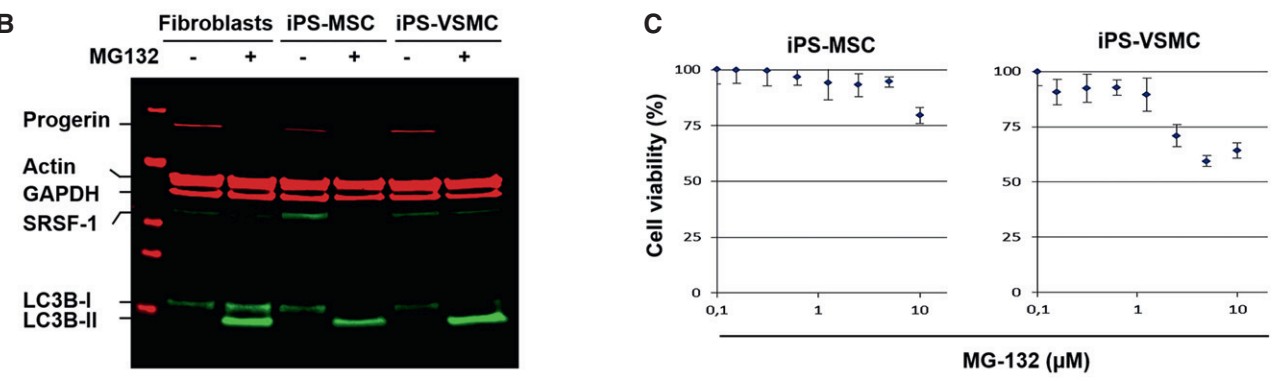

**Figure 6. MG132 reduces progerin and SRSF-1 levels in patient iPS-Derived MSC and VSMC.**

A  Immunofluorescence images of HGPS-derived iPS-VSMC and WT-derived iPS-VSMC cultured with DMEM medium containing either indicated MG132 concentrations (0.15, 0.31, 0.62, 1.25, 2.5, 5, and 10 μM) or DMSO (0) for 24 h and stained for progerin (green). Scale bar, 40 μm. (n = 3).

B  MG132 treatment resulted in a decrease of progerin levels. A representative Western blotting experiment in whole lysates showing progerin, actin, GAPDH, SRSF-1, LC3B-I, and LC3B-II expression in 5 μM MG132-treated HGPS fibroblasts (+), 2.5 μM MG132-treated HGPS iPS-MSC (+), and 1.2 μM MG132-treated HGPS iPS-VSMC (+) for 24 h, relative to DMSO-treated cells (−). (n = 4).

C  HGPS-derived iPS-MSC and iPS-VSMC viability was measured at 24 h post-treatment with MG132 at the indicated concentrations using CellTiter-Glo Luminescent Cell Viability Assay. Results are reported as viability percentage of MG132-treated cells relative to DMSO-treated cells. Results are expressed as mean ± SEM (n = 4).

Source data are available online for this figure.

cellular antiviral defense mechanisms using IFNs. Dramatic linear and rosette PML-NBs lacking substantial SUMO-1, Daxx, and Sp100 have been described as unique to early hESC cultures. These occur primarily between Day 0 and 2 of differentiation and become rare thereafter (Butler *et al*, 2009). PML bodies are larger in A-type lamin-deficient fibroblasts compared with their WT counterparts (Stixova *et al*, 2012).

We also showed that in HGPS patients' nuclei, PML-NBs contained, in addition to progerin, two nuclear envelope proteins: lamin B and emerin; conventional PML protein components were observed as follows: SP100, HP1, DAXX, CBP, and ATRX, while lamin A/C was absent. It has been shown that lamin A strongly interacts with the nucleoskeleton in comparison to progerin (Kubben *et al*, 2010). Thus, targeting of lamin A/C to PML-NBs could be inhibited by this interaction. On the opposite, localization of emerin into PML-NBs might result from the regulation of emerin localization by progerin, since there is a stronger affinity between emerin/progerin than emerin/lamin A and also a capacity of (ΔNLS) progerin to remove all endogenous emerin from the nuclear envelope (Wu *et al*, 2014). As well, lamin B1 has been shown to co-assemble preferentially with progerin rather than with wild-type lamin A in co-expression and FRET analyses (Delbarre *et al*, 2006), possibly explaining its colocalization with progerin into PML-NBs.

Because the 26S proteasome and ubiquitin colocalize within PML-NBs, we hypothesized that these structures might be involved in progerin degradation. MG132 was used in order to investigate how this drug, known to inhibit the proteasome, could influence the levels and localization of progerin. Unexpectedly, instead of increasing progerin levels, MG132 treatment resulted in a dramatic decrease of progerin. Our results suggested that the effect of MG132 and some of its analogs (MG115 and MG262) was mediated by another action rather than their known one on proteasome inhibition, since other analogs strongly inhibiting the proteasome activities (bortezomib and carfilzomib) were not capable of inducing effective progerin clearance. MG132, bortezomib and carfilzomib differ structurally, mechanistically and physicochemically (bortezomib: boronate class; carfilzomib: epoxyketones class; and MG132: aldehyde class), they also differ in their inhibition on the 3 proteasome activities and in their action reversibility (Britton *et al*, 2009; Kisselev *et al*, 2012). This indicates that these molecules could have different effects. In addition, it has been shown that bortezomib inhibits autophagy (Periyasamy-Thandavan *et al*, 2010; Kao *et al*, 2014), hence probably its inability to induce an effective progerin clearance compared to that of MG132. On the other hand, we did not observe a decrease of SRSF-1 levels in the presence of bortezomib or carfilzomib, in contrast to MG132 (data not shown).

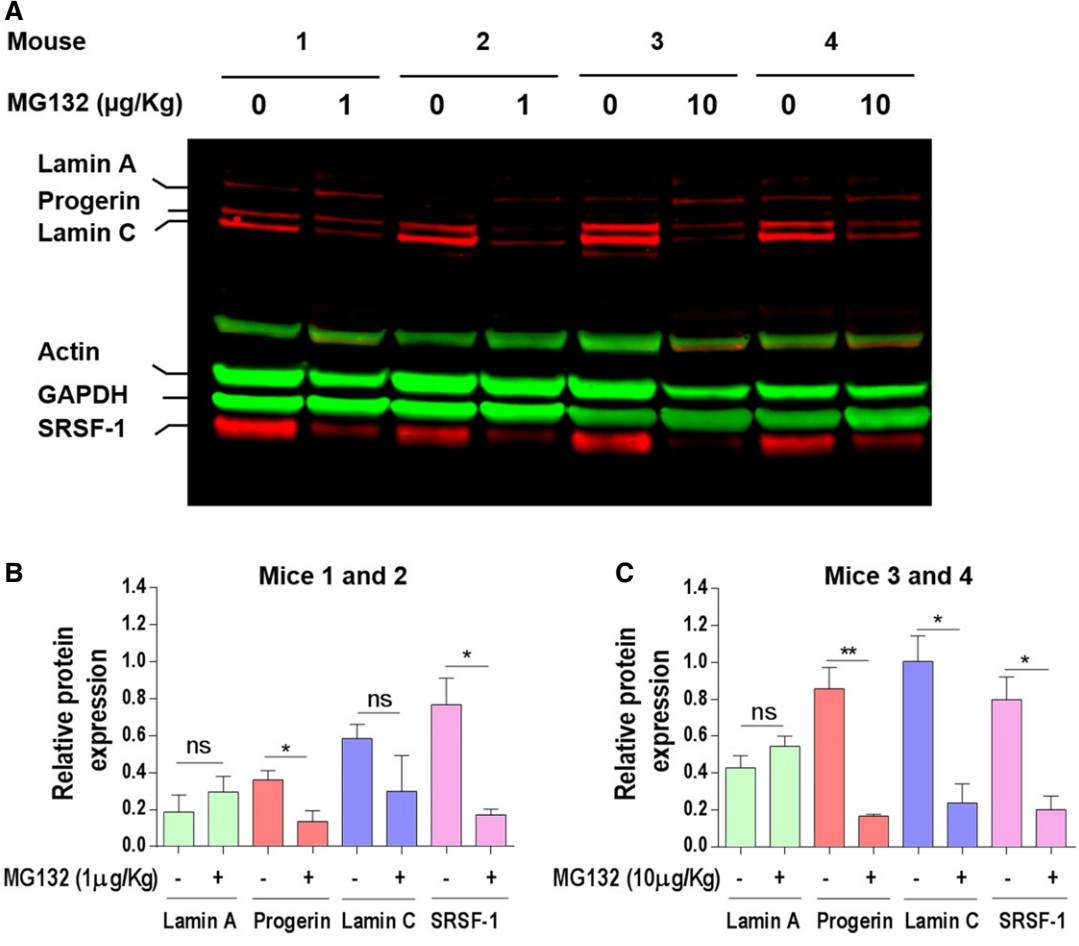

**Figure 7. MG132 reduces progerin and SRSF-1 levels in *Lmna*$^{G609G/G609G}$ mice muscle.**

A  Western blotting evaluation of lamin A/C, progerin, actin, GAPDH, and SRSF-1 in gastrocnemius muscle lysate of *Lmna*$^{G609G/G609G}$ mice treated with 1 µg/kg (mice 1 and 2) or 10 µg/kg (mice 3 and 4) MG132. The results are shown as four successive pairs (left: DMSO-treated (0 µg/kg) and right: MG132-treated (1 or 10 µg/kg) muscles, each corresponding to a single mouse).

B  Lamin A-, progerin-, lamin C-, and SRSF-1-specific bands, corresponding to DMSO-treated left muscle (−) and 1 µg/kg MG132-treated right muscle (+), were quantified by ImageJ software and their expression levels were normalized to GAPDH values.

C  Lamin A-, progerin-, lamin C-, and SRSF-1-specific bands, corresponding to DMSO-treated left muscle (−) and 10 µg/kg MG132-treated right muscle (+), were quantified by ImageJ software, and their expression levels were normalized to GAPDH values.

Data information: Results are expressed as mean ± SEM, $n = 5$, Student's *t*-test, *$P < 0.05$, **$P < 0.01$, MG132-treated vs. DMSO-treated mice muscle; the exact *P*-values are indicated in Appendix Table S1.

Interestingly, MG132 seemed to have a specific effect on reducing progerin *in vitro*, while sparing the other lamin isoforms, since the levels of lamin A and C were not affected but instead accumulated in treated cells; more generally, as expected, the global quantities of ubiquitinated proteins also increase in treated cells. By analyzing the levels and state of ubiquitination and sumoylation of progerin upon MG132 treatment under non-denaturing or denaturing Western blot conditions, we showed that when progerin is ubiquitinated, its levels are reduced upon treatment, while when it is both sumoylated and ubiquitinated, its levels are not changed. These results suggest for the first time that progerin is not degraded by similar means when it is sumoylated vs. when it is not. This idea is consistent with previous studies showing that both modifiers may be considered as having antagonistic effects on their substrates in a competitive manner (Anderson *et al*, 2012); it was also shown that

sumoylation may physiologically lead to subsequent ubiquitination and thus direct the target protein to the degradation processes (Geoffroy & Hay, 2009). Progerin sumoylation is consistent with its localization within PML-NBs, since the majority of proteins targeted to PML-NBs are sumoylated (Zhong *et al*, 2000). Moreover, this localization is consistent with the recognition and the clearance of misfolded proteins following the concerted action of PML, SUMO, and RNF4 (Gartner & Muller, 2014). Altogether, these observations suggest that progerin sumoylation would allow it to be addressed to PML-NBs, while its degradation requires its desumoylation and ubiquitination, probably due to competition between sumo and ubiquitin that could target the same residue as is the case of Ikappa Bα or MDM2 (Muller *et al*, 2001).

We showed for the first time that upon MG132 treatment in HGPS cell lines, progerin is delocalized to the nucleolus, where it

mostly localizes after 24 h treatment. Additionally, our results showed that not only 26S and p53 proteins localized to the nucleolus upon treatment, consistent with previous data reporting that MG132 can lead to the translocation of PML-NBs components into the nucleolus (Mattsson et al, 2001), but also emerin and, to a lesser extent, lamin B suggesting that nucleolus might function as an alternative sequestration site that immobilizes and temporarily stores some PML-NBs components including accumulated progerin when proteasome-dependent degradation is blocked (Visintin & Amon, 2000). Interestingly, we showed that LC3B, a macroautophagy marker, undergoes the same dynamics as progerin upon MG132 treatment.

Previous data indicate that DOR (Diabetes- and Obesity-related protein), a PML-associated protein, which positively regulates stress-induced autophagy, undergoes nucleo-cytoplasmic shuttling involving a transition through the nucleolus (Mauvezin et al, 2012) and mediates the autophagic export of nuclear LC3B (Huang et al, 2015). p62, a cargo receptor for autophagic degradation of ubiquitinated proteins also localizes in PML-NBs and shuttles continuously between nuclear and cytosolic compartments at a high rate (Pankiv et al, 2010) and PML bodies themselves can be dynamically exchanged between the nucleus and cytoplasm (Dieriks et al, 2011; Houben et al, 2013). The mechanism of the export into cytosol of nuclear protein aggregates allowing their degradation by macroautophagy remains to be elucidated (Isakson et al, 2013). In line with our findings, showing that exportin-1 dependent nuclear export through the nuclear pores was not required for the MG132-mediated clearance of progerin, it has been shown that lamin B1 degradation is also achieved by nucleus-to-cytoplasm transport that delivers lamin B1 to the lysosome without passing through the nuclear pores. Indeed, nuclear materials to be degraded will be encapsulated through a process similar to exocytosis (Dou et al, 2015; Boban & Foisner, 2016; King & Lusk, 2016; Luo et al, 2016).

These results suggest that proteins localized within PML-NBs may also undergo the same behavior. Indeed, upon MG132 treatment, progerin could be observed in the cytoplasm, inside vacuoles co-stained with several autophagic markers as p62, LAMP-2 and LC3B.

These results show for the first time that progerin, a permanently prenylated protein which was thought to be stably localized in nuclei, due to its altered anchoring to membranes (Dechat et al, 2007) and its heterodimerization with wild-type lamin (Delbarre et al, 2006), can be forced out of the nuclear compartment, where it is known to exert its toxic effects.

One of the major findings of this study is the demonstration that proteasome inhibition with specific molecules induces partial progerin clearance by activation of autophagy. It is indeed known that impairment of the UPS is compensated by activation of autophagy (Zhu et al, 2010; Zang et al, 2012; Wang et al, 2013a; Tang et al, 2014).

This activation in HGPS cells is supported by the increased amounts of LC3B-II on immunoblotting assays, progerin delocalization into cytoplasmic autophagic vacuoles, increased autophagic transcript levels using RNA-seq experiments and partial restoration of progerin levels in presence of chloroquine or bafilomycin A1 which are commonly used as autophagy inhibitors. Finally, we could show that, upon MG132 treatment, the blockade of both autophagy and caspase-mediated cleavage of the progerin splicing regulator

SRSF-1, allowed the complete restoration of basal progerin levels, confirming the dual action of MG132 leading to progerin clearance. Consistent with our results, the caspase-mediated reduction of SRSF-1 levels had already been described upon MG132 stimulation (Moulton et al, 2014). SRSF-5 accumulation upon MG132 treatment had also been reported, since SRSF5 proteolysis is mediated by the proteasome (Breig & Baklouti, 2013). Therefore, SRFS-5 appears to be involved only for a short time after treatment, hence the possibility of restoring the basal progerin levels only by inhibiting caspase-dependent cleavage of SRSF-1 and autophagy when cells are exposed to MG132. The dynamics of progerin localizations and putative involvement of the three main protein degradation systems in progerin clearance upon MG132 treatment are modeled in Fig EV5. In summary, in primary cultures of fibroblasts from HGPS patients, progerin is sequestered in PML-NBs. Treatment of these cells with MG132 initially induces a caspase-dependent reduction of SRSF-1 levels as well as a rapid but not sustained increase of SRSF-5 levels even after the treatment renewal. The modulation of the levels of these two splicing factors is in favor of a decrease in progerin levels. In parallel, MG132 at first induces nucleolar translocation of progerin and then its accumulation and degradation in autophagic vacuoles.

Since SRSF-1 and SRSF-5 are involved in the control of alternative lamin A/progerin splicing, we investigated lamin A/C expression levels upon MG132 treatment and found that although lamin A and to a lesser extent lamin C transcripts were reduced. The reduction of lamin A transcripts is not surprising given that the lamina A 5'SS is also recognized by SRSF-1 independently of the HGPS mutation that promotes the reinforcement of another weak 5' splice site also recognized by SRSF-1 and leading to progerin production. On the contrary, lamin A/C protein levels were not reduced; the poor correlation between transcript and protein levels, which is observed in other contexts (Maier et al, 2009), is probably due to multiple factors which involve the transcriptional and translational regulation steps and/or the differential half-lives of proteins and mRNAs under MG132 treatment.

Notably, we also showed that progerin reduction mediated by MG132 significantly improved nuclear shape abnormalities, reduced the number of senescent cells, increased the number of viable and proliferating cells, restored the normal cellular phenotype, and improved the expression levels of misregulated gene in HGPS cell lines. In addition, the ability of this treatment to reduce progerin and SRSF-1 levels was validated in two additional cell models: HGPS-derived iPS-MSC and VSMC, the latter being at the crossroads of HGPS cardiovascular pathophysiology.

MG132 was reported to have a significant preventive and therapeutic effect on cardiovascular and renal injury (Chatterjee et al, 2012; Wang et al, 2013b) and seems to be a potentially effective drug in the prevention of oxidative damage (Dreger et al, 2010; Miao et al, 2013) and of accelerated atherosclerosis in rabbits (Feng et al, 2010). Interestingly, these features are exhibited by HGPS patients who might benefit from the same treatment, independently from its specific action on progerin levels. All these results indicate that MG132 can have beneficial effects in vivo without exerting toxic effects.

Variable levels of progerin reduction were observed upon systemic MG132 treatment suggesting that the molecule is unstable when injected systemically in $Lmna^{G609G/G609G}$ mice; indeed, it has

been shown that MG132 is rapidly metabolized by hepatic CYP3A, becoming ineffective (Lee *et al*, 2010; Mlynarczuk-Bialy *et al*, 2014). This is the reason why we have subsequently performed intramuscular injections so that the molecule could act locally (one hindlimb treated vs. the contralateral hindlimb injected with vehicle, as an intra-individual control, to avoid any bias due to inter-individual variations of baseline progerin levels). Therefore, systemic injections of MG132 in *Lmna*$^{G609G/G609G}$ mice, which of course will be needed to target the different organs involved in the pathophysiology of progeria, will likely require the setting up of an appropriate galenic form of this short peptide in order to increase its half-life. *In vivo,* MG132 injection resulted in a significant decrease of progerin and SRSF-1. Only at high dose, MG132 treatment induced a decrease in lamin C. However, the levels of lamin A remained constant or increased slightly, it is important to note that decrease in lamin C levels has no deleterious effect on mice as previously shown in *Lmna*$^{LAO/LAO}$ mice which were healthy despite a complete absence of lamin C (Coffinier *et al*, 2010).

The results reported herein provide the basis for the development of a novel promising drug for children affected with progeria, based on impressive progerin clearance *in vitro* and *in vivo* upon treatment by MG132. Considering the activity of MG132 on SRSF-1 downregulation, this molecule may have beneficial effects on diseases involving the altered splicing driven by SRSF-1 or SRSF-1 overexpression (Karni *et al*, 2007). Finally, the findings described in this work may offer insights into mechanisms of physiological aging, during which progerin has also been shown to play a role (Scaffidi & Misteli, 2006; McClintock *et al*, 2007).

# Materials and Methods

## Cell culture

Human dermal fibroblasts (fibroblasts established from a skin biopsy) from five control subjects and five patients who carried the HGPS p.Gly608Gly mutation were obtained from the Coriell Cell Repository. Cells were cultured in Dulbecco's modified Eagle's medium (Life Technologies) supplemented with 15% fetal bovine serum (Life Technologies), 2 mM L-glutamine (Life Technologies), and 1× penicillin–streptomycin (Life Technologies) at 37°C in a humidified atmosphere containing 5% $CO_2$. Testing for mycoplasma contamination was performed regularly. Fibroblasts were cultured in the presence or absence of MG132 (474790, Merck Chemical LTD), MG115 (SCP0005, Sigma), MG262 (I-120-200, R&D Systems), bortezomib (S1013, Euromedex), carfilzomib (S2853, Euromedex), chloroquine diphosphate crystalline (C6628, Sigma), bafilomycin A1 (B1793, Sigma), caspase-6 inhibitor Z-VEID-FMK (FMK006, R&D Systems), pan-caspase inhibitor Z-VAD-FMK (FMK001, R&D Systems), leptomycin B (L2913, Sigma), 2-D08 (SML1052, Sigma), and ginkgolic acid C15:1 (74741, Sigma). The experiments were performed on fibroblasts of HGPS patients and healthy subjects matched for age and passage.

## Quantitation of abnormal nuclear morphology

Fibroblasts from three HGPS patients and three normal individuals were cultured with DMEM medium containing 5 μM MG132,

or the same volume of vehicle (DMSO, 0.025% v/v) for 48 h. Cells stained for lamin A/C or DAPI were examined by fluorescence microscopy with an Axioplan 2 imaging microscope (Carl Zeiss). The percentage of normal nuclei (nuclei with a smooth oval shape) and abnormal nuclei (nuclei with blebs, irregular shape, or multiple folds) was calculated using the Nuclear Irregularity Index (NII) plugin of the ImageJ software (Version 1.6.0, NIH, USA). The analysis uses measures of nuclear area and of four parameters of irregularity, named aspect, area/box, radius ratio, and roundness. These four parameters are used to generate a Nuclear Irregularity Index (NII) which, added to area measurement, classify the nuclei in normal or abnormal morphology. At least 200 fibroblast nuclei were randomly selected for each cell line and examined in three independent experiments. Results are expressed graphically as the average percentage of the total nuclei counted.

## Quantitation of abnormal PML-NBs morphology

The percentage of abnormal PML-NBs (ring-like and thread-like) in fibroblasts from four HGPS patients and four normal individuals matched for age and passage was calculated by two independent observers using a manual blind counting. At least 200 fibroblast nuclei were randomly selected for each cell line ($n = 4$) and examined. Results are expressed graphically as the average percentage of the total nuclei counted.

## Proteasome activities assays

The proteasome activities were determined in HGPS and control cells by using the Proteasome-Glo™ 3-Substrate Cell-Based Assay System (Promega) that allows to measure the chymotrypsin-like, trypsin-like, or caspase-like protease activity associated with the proteasome complex in cultured cells. The Proteasome-Glo™ Cell-Based Reagents each contain a specific luminogenic proteasome substrate. These peptide substrates are Suc-LLVY amino-luciferin (succinyl-leucine-leucine-valine-tyrosine-aminoluciferin), Z-LRR- aminoluciferin (Z-leucine-arginine-arginine-aminoluciferin) and Z-nLPnLD-aminoluciferin (Z-norleucine-proline-norleucine-aspartate-aminoluciferin) for the chymotrypsin-like, trypsin-like and caspase-like activities, respectively. Cells (10,000 cells/well) were cultured in 100 μl/well in a 96-well plate at 37°C, 5% $CO_2$. The plate was allowed to equilibrate to 22°C before 100 μl/well of substrate and luciferin detection reagent was added. The contents of the wells were mixed at 700 rpm using a plate shaker for 2 min and incubated at room temperature for 10 min. Following cleavage by the proteasome, the substrate for luciferase (aminoluciferin) is released, allowing the luciferase reaction to proceed and produce light. Luminescence is determined as relative light units (RLUs) using a GloMax-Multi Detection System: Luminometer (Promega, USA).

## Measurement of senescence

Senescence was measured with a Beta-Glo Assay kit (Promega), according to the manufacturer's instructions. Luminescence intensity was determined as RLUs using a GloMax-Multi Detection System: Luminometer (Promega, USA).

## Viability and cytotoxicity assays

Viability and cytotoxicity were measured with a MultiTox-Fluor Multiplex Cytotoxicity Assay (Promega), according to the manufacturer's instructions. The MultiTox-Fluor Assay simultaneously measures two protease activities: One is a marker of cell viability using a fluorogenic, cell-permeant peptide substrate (glycyl-phenylalanylamino fluorocoumarin; GF-AFC), and the other is a marker of cytotoxicity (bisalanyl-alanyl-phenylalanyl-rhodamine 110; bis-AAF-R110). The live- and dead-cell proteases produce different products, AFC and R110, which have different excitation and emission spectra, allowing them to be detected simultaneously. Results are provided in relative fluorescence units (RFU) measured using a GloMax-Multi Detection System: Luminometer (Promega, USA). Data are representative of at least three independent experiments done in triplicate.

## Proliferation assay

Cells proliferation rate was measured either with a CellTiter 96 Aqueous One Solution Cell Proliferation Assay (Promega) or with a BrdU Cell Proliferation ELISA Kit (Abcam), according to the manufacturer's instructions. Absorbance was monitored with a GloMax-Multi Detection System: Luminometer (Promega, USA).

## Fibroblasts reprogramming

Generation of induced pluripotent stem cells (iPS) by reprogramming was carried out using fibroblasts isolated from HGPS patient biopsies provided by Coriell Institute (Camden, USA) or performed in the Assistance Publique Hôpitaux de Marseille and issued from our Biological Ressources Center, which declared its collections of human biological samples to the French ministry of health (declaration number DC-2008-429) and was authorized from the French ministry of health to cede them for research studies (authorization number AC-2011-1312). Informed consents were obtained from the patients or the parents of minor patients included in this work. Experiments are conformed to the principles set out in the WMA Declaration of Helsinki and the Department of Health and Human Services Belmont Report.

Cell lines were reprogrammed to iPSC using Yamanaka's original method based on retrovirus-mediated transfection of four transcription factors, namely Oct3/4, Sox2, c-Myc, and Klf4 (Takahashi *et al*, 2007). The iPSC lines were amplified up to the 15th passage before differentiation.

## Pluripotent stem cells culture and differentiation

HGPS iPSC were grown on STO mouse fibroblasts, inactivated with 10 mg/ml mitomycin C, seeded at 30,000/cm$^2$, and grown as previously described (Aubry *et al*, 2008). For differentiation, iPSC were differentiated into mesenchymal stem cells (MSC-iPSC) using directed protocols for differentiation previously published (Guenou *et al*, 2009; Giraud-Triboult *et al*, 2011; Nissan *et al*, 2011). VSMC were obtained from iPSC by fluorescent-activated cell sorting (FACS) separation of CD31$^+$ cells (Levenberg *et al*, 2010).

## Mice

Knock-in mouse model (*Lmna$^{G609G/G609G}$*) carrying the c.1827C>T (p.Gly609Gly) mutation in the endogenous *Lmna* gene, corresponding to the human HGPS mutation c.1824C>T (p.Gly609Gly), has been described previously (Osorio *et al*, 2011). MG132 (1 or 10 μg/kg body weight) was injected under isoflurane anesthesia into the right gastrocnemius muscle of 3-month-old male and female homozygous knock-in mice (*Lmna$^{G609G/G609G}$*) three times per week for 2 weeks. Contralateral muscle served as control (DMSO). MG132 was dissolved in dimethyl sulphoxide (DMSO) and then diluted in 100 μl sterile phosphate-buffered saline (PBS) for intramuscular injection. Neither vehicle alone nor MG132 treatment produced any apparent damage or stress responses in mice.

## siRNA transfection

For knockdown of SRSF-1, HGPS cells were transfected with 10 nM siRNA using INTERFERin (Polyplus Transfection). HGPS cells were seeded the day before transfection at 100,000–200,000 cells in 6-well culture vessel. 22 pmoles of siRNA was diluted in 200 μl of serum-free DMEM medium, mixed with 8 μl of INTERFERin reagent. Tubes were vortexed 10 s and incubated 10 min at room temperature. The transfection mix was added to the cells in serum containing DMED medium. Medium was changed after 4 h and cells incubated at 37°C, 5% CO$_2$ for 48 h. SRSF-1 siRNA (AM16708) was purchased from Thermo Fisher Scientific.

## Delivery of morpholino oligonucleotides

For morpholino delivery, we used the Endoporter system and followed the manufacturer's instructions (Gene Tools, LLC, Philomath, OR, USA). Briefly, cells were plated at 50% subconfluence in DMEM (Life Technologies) containing 10% fetal bovine serum (Life Technologies) and 2 mm L-glutamine (Life Technologies) at 37°C in a humidified atmosphere of 5% CO$_2$. Each morpholino oligonucleotide was added at a final concentration of 20 μM to cell cultures in six-well plates. Endoporter reagent was added at a final concentration of 6 μM. Cells were retransfected every 48 h. All transfection experiments were repeated three times. The morpholino oligonucleotides used in this study were as follows: Ex10 (5′-GCTACCACT CACGTGGTGGTGATGG-3′) that bound to the exon 10-lamin A splice donor site and Ex11 (5′-GGGTCCACCCACCTGGGCTCCTGAG-3′) that bound the c.1824C>T; p.Gly609Gly-mutated sequence in the region of the LMNA transcript. Control scrambled: Ex10-scrambled 5′-ATCGGCTTGTCGCGTGAGCGATCGA-3′ and Ex11-scrambled 5′-ACCAGTGGCGTCGCCTCGCAGGTCC-3′.

## Antibodies

Antibodies used in the study included: a rabbit anti-lamin A/C polyclonal antibody which reacts with lamin A, lamin C, and progerin (SC-20681, used at 1:1,000 dilution for the Western blot analyses and at 1:200 for immunofluorescence labeling, Santa Cruz Biotechnology, Inc.); a mouse anti-progerin monoclonal antibody (sc-81611 used at 1:1,000 dilution for the Western blot analyses and at 1:200 for immunofluorescence labeling, Santa Cruz Biotechnology, Inc.); a rabbit anti-PML polyclonal antibody (sc-9863-R used at 1:200 for

immunofluorescence labeling, Santa Cruz Biotechnology, Inc.); a mouse anti-PML monoclonal antibody (sc-966 used at 1:100 for immunofluorescence labeling, Santa Cruz Biotechnology, Inc.); a goat anti-SP100 polyclonal antibody (sc-16328 used at 1:50 for immunofluorescence labeling, Santa Cruz Biotechnology, Inc.); a mouse anti-HP1 monoclonal antibody (2HP-1H5-AS used at 1:100 for immunofluorescence labeling, Euromedex); a rabbit anti-ATRX polyclonal antibody (sc-15408 used at 1:50 for immunofluorescence labeling, Santa Cruz Biotechnology, Inc.); anti-DAXX (ab32140 used at 1:50 for immunofluorescence labeling, Abcam); a rabbit polyclonal anti-CBP antibody (06-297 used at 1:200 for immunofluorescence labeling, Upstate); a mouse anti-ubiquitin antibody (PW8805 used at 1:10 for immunofluorescence labeling, Enzo); a mouse anti-mono- and polyubiquitinylated conjugates monoclonal antibody (FK2, used at 1:1,000 for the Western blot analyses, Enzo); a rabbit anti-fibrillarin polyclonal antibody (ab5821 used at 1:100 for immunofluorescence labeling, Abcam); a goat polyclonal anti-lamin B1/B2 antibody (sc6217 used at 1:100 for immunofluorescence labeling, Santa Cruz Biotechnology, Inc.); a goat anti-emerin polyclonal antibody (sc-8086 used at 1:100 for immunofluorescence labeling, Santa Cruz Biotechnology, Inc.); a mouse anti-26S monoclonal antibody (65144 used at 1:1 for immunofluorescence labeling, Progen Biotechnic); a mouse anti-Nup153 monoclonal antibody (ab93310, used at 1:10 for immunofluorescence labeling, Abcam); a mouse anti-p53 monoclonal antibody (ab26, used at 1:200 for immunofluorescence labeling, Abcam); a rabbit anti-p62 polyclonal antibody (ab37024, used at 1:1,000 for the Western blot analyses and at 1:100 for immunofluorescence labeling, Abcam); a rabbit anti-LC3B polyclonal antibody (#2775, used at 1:1,000 for the Western blot analyses and at 1:400 for immunofluorescence labeling, Cell Signaling); a rabbit anti-SQSTM1/p62 polyclonal antibody (#5114, used at 1:1,000 for the Western blot analyses and at 1:100 for immunofluorescence labeling, Cell Signaling); a rabbit anti-lamin C polyclonal antibody (BP4505, used at 1:200 for immunofluorescence labeling, Acris Antibodies); a rabbit anti SRSF1 monoclonal antibody (ab129108, used at 1:1,000 for the Western blot analyses, Abcam); a mouse anti-glyceraldehyde-3-phosphate dehydrogenase monoclonal antibody (MAB374, used at 1:10,000 for the Western blot analyses, Merck Millipore); a mouse anti-α-tubulin monoclonal antibody (T6074, used at 1:10,000 for the Western blot analyses, Sigma) or a mouse anti-β-actin monoclonal antibody (AC-40, used at 1:10,000 for the Western blot analyses, Sigma); a rabbit anti-Sumo2/3 polyclonal antibody (ab-3742, used at 1:1,000 for the Western blot analyses, Abcam); a rabbit anti-SRSF-5 polyclonal antibody (PA5-13571, used at 1:1,000 for Western blot analyses, Thermo Fisher Scientific); a rabbit anti-γH2A.X (phospho S139) polyclonal antibody (ab2893, used at 1:200 for immunofluorescence labeling, Abcam); a rabbit anti-histone H3 (tri-methyl K9) polyclonal antibody (ab8898, used at 1:100 for immunofluorescence labeling, Abcam); a rabbit anti-LAP2α polyclonal antibody (ab5162, used at 1:100 for immunofluorescence labeling, Abcam); a rabbit anti-lamin-B1 polyclonal antibody (ab 16048, used at 1:100 for immunofluorescence labeling, Abcam); a goat anti-HP1α polyclonal antibody, used at 1:100 for immunofluorescence labeling, Abcam); and a mouse anti-calreticulin monoclonal antibody (H00000811-M01, used at 1:50 for immunofluorescence labeling; Novus Biologicals).

**Fluorescence microscopy**

Cells collected by centrifugation were spread onto polylysine-coated slides by centrifugation (300 rpm for 5 min) with a cytospin (Shandon). Cells were then fixed at room temperature for 10 min in 2% (w/v) paraformaldehyde in PBS pH 7.2. Slides were stored at −80°C prior to use. Cells were permeabilized using 100 μl permeabilization buffer (0.5% Triton X-100, 50 mM NaCl, 300 mM sucrose, 20 mM HEPES pH 7.5, 3 mM MgCl2) for 3 min at RT. Non-specific antibody binding was blocked with 3% bovine serum albumin in PBS (w/v) (PBS-BSA) for 30 min. The permeabilized cells were incubated with the primary antibodies for 3 h at 37°C. After washing, the cells were then incubated with secondary antibodies (A11001, A11058, Life Technologies; 1/400) for 20 min at 37°C. Nuclei were stained with DAPI (50 ng/ml) diluted in Vectashield (Abcys) for 10 min at RT. Samples were fixed in 4% paraformaldehyde for 5 min. The stained cells were observed either on an Axioplan 2 imaging microscope (Carl Zeiss) or examined through a Nikon inverted microscope attached to a laser confocal scanning system (Leica). All antibodies were tested in individual staining reactions for their specificity and performance. Controls without primary antibody were all negative.

**RNA isolation, reverse transcription, and real-time PCR**

Total RNA was isolated using the RNeasy plus extraction kit (Qiagen, Valencia, CA, USA). 1 μg of RNA was reverse transcribed using the High Capacity cDNA Reverse Transcription Kit (Applied Biosystems). Real-time PCR amplification was carried out with the TaqMan® Gene Expression Master Mix (Applied Biosystems) on a LightCycler 480 (Roche, Germany) using predesigned primers for GAPDH (Hs00266705_g1), progerin (F: ACTGCAGCAGCTCGGGG. R: TCTGGGGGCTCTGGGC and probe: CGCTGAGTACAACCT), lamin A (F: TCTTCTGCCTCCAGTGTCACG. R: AGTTCTGGGGGCTC TGGGT and probe: ACTCGCAGCTACCG), and lamin C (F: CAACTC CACTGGGGAAGAAGTG. R: CGGCGGCTACCACTCAC and probe: ATGCGCAAGCTGGTG) (Applied Biosystems) using the program: UNG incubation at 50°C for 2 min, initial denaturation at 95°C for 10 min, 40 cycles of amplification: denaturation at 95°C for 15 s and annealing at 60°C for 1 min. All PCRs were performed in triplicate. Threshold cycle (Ct) values were used to calculate relative mRNA expression by the $2^{-\Delta\Delta C_T}$ relative quantification method with normalization to GAPDH expression.

**RNA sequencing (ArrayExpress accession number: E-MTAB-5807)**

RNA sequencing was performed by IntegraGen (Evry, France). RNA samples were used to generate sequencing libraries with the TruSeq Stranded mRNA Sample Prep' Illumina®. The libraries were sequenced on an Illumina HiSeq 4000 sequencer, yielding approximately 35 million 2 × 75-bp paired-end reads.

*Quality control*
Quality of reads was assessed for each sample using FastQC (http://www.bioinformatics.babraham.ac.uk/projects/fastqc/).

## Sequence alignment and quantification of gene expression

A subset of 500,000 reads from each Fastq file was aligned to the reference human genome hg38 with TopHat2 to determine insert sizes with Picard. Full Fastq files were aligned to the reference human genome hg38 with TopHat2 ($-p$ 24 $-r$ 150 $-g$ 2 –library-type fr-firststrand). Reads mapping to multiple locations removed. Gene expression was quantified using two non-overlapping transcriptome annotations: the full Gencode v25 annotation as well as a complementary lncRNA annotation. To obtain this second annotation, exhaustive lncRNA annotations were combined from Cabili *et al* and Iyer *et al* and lncRNAs not represented in the Gencode database were selected. HTSeq was used to obtain the number of reads associated to each gene in the Gencode v25 database (restricted to protein-coding genes, antisense and lincRNAs) and to each gene in the additional lncRNA database. The Bioconductor DESeq package was used to import raw HTSeq counts for each sample into R statistical software and extract the count matrix. After normalizing for library size, the count matrix was normalized by the coding length of genes to compute FPKM scores (number of fragments per kilobase of exon model and millions of mapped reads). Bigwig visualization files were generated using the bam2wig python script.

## Unsupervised analysis

The Bioconductor DESeq package was used to import raw HTSeq counts into R statistical software, to obtain size factors, and to calculate a variance stabilizing transformation (VST) from the fitted dispersion–mean relations to normalize the count data. The normalized expression matrix from the 1,000 most variant genes (based on standard deviation) was used to classify the samples according to their gene expression patterns using principal component analysis (PCA) and hierarchical clustering.

## Differential expression analysis

The Bioconductor DESeq package was used to import raw HTSeq counts into R statistical software, to obtain size factors and dispersion estimates and to test differential expression. Only genes expressed in at least one sample (FPKM $\geq$ 0.1) were tested to improve the statistical power of the analysis. A q-value threshold of $\leq$ 0.05 was applied to define differentially expressed genes.

## Western blot

Total fibroblast proteins were extracted in 200 μl of NP40 Cell Lysis Buffer (Invitrogen, Carlsbad, CA, USA) containing Protease and Phosphatase Inhibitor Cocktail (Thermo Scientific). Alternatively, cells were lysed with urea [8 M urea, 5 mM dithiothreitol, 150 mM NaCl, 50 mM Tris–Cl pH 7.5, Protease and phosphatase Inhibitor Cocktail (Thermo Scientific)]. Cells were sonicated twice (30 s each), incubated at 4°C for 30 min and then centrifuged at 10,000 *g* for 10 min. Protein concentration was evaluated with the bicinchoninic acid technique (Pierce BCA Protein Assay Kit), absorbance at 562 nm is measured using nanodrop 1000 (Thermo Fisher Scientific) Equal amounts of proteins (40 μg) were loaded onto 10% Tris-glycine gel (CriterionTM XT precast gel) using XT Tricine Running Buffer (Bio-Rad, USA). After electrophoresis, gels were electro transferred onto nitrocellulose membranes or Immobilon-FL polyvinylidene fluoride membranes (Millipore), blocked in odyssey Blocking Buffer diluted 1:1 in PBS for 1 h at room temperature, and incubated overnight at 4°C or 2 h at room temperature with various primary antibodies. Blots were washed with TBS-T buffer [20 mM tris (pH 7.4), 150 mM NaCl, and 0.05% Tween 20] and incubated with 1:10,000 IR-Dye 800-conjugated secondary donkey anti-goat or IR-Dye 700-conjugated secondary anti-mouse antibodies (LI-COR Biosciences) in odyssey blocking buffer (LI-COR Biosciences). For IR-Dye 800 and IR-Dye 700 detection, an odyssey Infrared Imaging System (LI-COR Biosciences) was used. GAPDH, α-tubulin, or β-actin were used as a total cellular protein loading control.

## Immunoprecipitation

Immunoprecipitation was carried out using the Dynabeads Protein G Immunoprecipitation kit (Invitrogen). 5 μg of mouse monoclonal anti-progerin antibody (Abcam) was added to 50 μl (1.5 mg) Dynabeads and rotated at room temperature for 10 min. The supernatant was removed and beads–antibodies complex washed in 200 μl PBS with Tween 20 containing 5 mM N-ethylmaleimide (NEM, Thermo scientific). The supernatant was removed, and Dynabeads were incubated with 200 μl cell lysate (200 μg protein) extracted using NP40 Cell Lysis Buffer (Invitrogen) containing protease, Phosphatase Inhibitor Cocktail (Thermo Scientific), and 5 mM NEM, and the tubes were rotated at room temperature for 30 min. The beads were washed three times according to the manufacturer's protocol. Proteins were eluted with 20 μl elution buffer containing 5 mM NEM. SDS–PAGE and immunoblotting were performed as described above.

## Statistics

Statistical analyses were performed with the GraphPad Prism software. Differences between groups were assayed using a two-tailed Student's *t*-test using GraphPad Prism. In all cases, the experimental data were assumed to fulfill *t*-test requirements (normal distribution and similar variance); in those cases, where the assumption of the *t*-test was not valid, a nonparametric statistical method was used (Mann–Whitney test). A *P*-value less than 0.05 was considered as significant. Error bars indicate the standard error of the mean.

## Study approval

Animal experiments have been carried out in compliance with the ARRIVE (Animal Research: Reporting of *in vivo* Experiments) guidelines and the European guidelines for the care and use of laboratory animals (EU directive 2010/63/EU) and in accordance with the recommendations provided by the guide for the care and use of the laboratory animals of the French National Institute for Health and Medical Research (INSERM) and the ethical committee of Aix-Marseille University.

**Expanded View** for this article is available online.

## Acknowledgements

We thank P. Roubertoux, N. Dasilva, F. Merono, C. Bartoli, and Anne Laure Egesipe for their support and assistance. We acknowledge the

### The paper explained

#### Problem

Hutchinson–Gilford progeria syndrome (HGPS) is a lethal premature and accelerated aging disease which affects one in 4–8 million children. Our team identified the recurrent mutation causing this disease in the LMNA gene, encoding lamin A/C (De Sandre-Giovannoli *et al*, 2003). Progerin, a toxic lamin A derivative issued from aberrant prelamin A mRNA splicing, is an abnormally farnesylated and truncated protein that accumulates in patients' nuclei, exerting multiple toxic effects. We have explored the mechanisms of accumulation and of possible degradation of progerin and identified a very promising new therapeutic avenue.

#### Results

We show that progerin is sequestered into abnormal promyelocytic leukemia nuclear bodies (PML-NB). We found that the proteasome inhibitor MG132 induces progerin degradation through translocation into autophagic vacuoles, involving a previous passage through the nucleolus. MG132 also strongly reduces progerin production through downregulation of SRSF-1 and accumulation of SRSF-5, controlling prelamin A mRNA aberrant splicing. We also report here that MG132 has beneficial effects, reversing cellular pathological phenotypes in HGPS fibroblasts and decreasing progerin and SRSF-1 levels in iPSC-derived MSC and VSMC from HGPS patients' fibroblasts. In the *Lmna*$^{G609G/G609G}$ mouse model of progeria, local intramuscular injections of MG132 lead as well to downregulation of SRSF-1 and progerin clearance.

#### Impact

We demonstrate progerin reduction based on MG132 dual action and shed light on a promising class of molecules toward a potential therapy for children with HGPS. Interestingly, MG132 belongs to a class of proteasome inhibitors, including MG115 and MG262, which have similar effect on progerin clearance, as shown in HGPS fibroblasts and iPS cells lineages used during this study. A new promising path thus opens for the clinical development of this novel class of molecules, toward regulatory approval and future clinical trials for patients affected with progeria and related disorders.

support of Andrée Robaglia-Schlupp and Karine Bertaux for technical support. The authors are grateful to Marc Bartoli and Marc Peschanski for critical discussions and their support throughout these studies and Carlos López-Otín for critical discussions, reading, and comments on this work. This work was supported by grants from Institut National de la Santé et de la Recherche Médicale, Aix-Marseille University, A*Midex Foundation (Program VinTAGE) and the Association Française contre les Myopathies (AFM grant MNH-Decrypt 2011-2015 and TRIM-RD 2016-2020 to NL). This study is part of the FHU A*MIDEX project MARCHE n.ANR-11-IDEX-001-02 funded by the "Investissement d'avenir" French government program, managed by the French National Research Agency (ANR).

### Author contributions

KH, DD, and XN performed the experiments. KH, CN, and NL analyzed the data. NL, ADSG, PC, and MGM were responsible for designing and supervising the project. KH, ADSG, and NL conceived, designed the experiments, and wrote the manuscript. KH and NL edited the manuscript.

### Conflict of interest

The authors declare that they have no conflict of interest.

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
