## [Review Process File · EMBO Molecular Medicine]

MG132-induced progerin clearance is mediated by autophagy activation and splicing regulation

Karim Harhour, Claire Navarro, Danielle Depetris, Marie-Geneviève Mattei, Xavier Nissan, Pierre Cau, Annachiara De Sandre-Giovannoli, Nicolas Lévy

Corresponding author: Nicolas Lévy, Aix Marseille University

Review timeline:

Submission date:	08 November 2016
Editorial Decision:	22 December 2016
Revision received:	21 April 2017
Editorial Decision:	19 May 2017
Revision received:	02 June 2017
Accepted:	07 June 2017

Transaction Report:

Editor: Céline Carret

1st Editorial Decision

22 December 2016

Thank you for the submission of your manuscript to EMBO Molecular Medicine. We have now heard back from the three referees whom we asked to evaluate your manuscript. Although the referees find the study to be of potential interest, they also raise a number of concerns that need to be addressed in the next version of your article.

You will see from the comments below that all three referees find the study interesting, with potentially important findings that could bear clinical relevance. Referees 1 and 3 are concerned about the limited provision of mechanistic insights but also translational aspects and recommend developing both by providing more data into MG132 treatment in vitro and in vivo for example. Referee 2 noticed in several places that controls are missing and reproducibility of the findings are unclear and this must absolutely be improved. Finally, all insist that the paper should be refocused and suggest rewriting a tighter discussion.

Given the balance of these evaluations, we feel that we can consider a revision of your manuscript if you can address the issues that have been raised within the space and time constraints outlined below. Please note that it is EMBO Molecular Medicine policy to allow only a single round of revision and that, as acceptance or rejection of the manuscript will depend on another round of review, your responses should be as complete as possible.

I look forward to receiving your revised manuscript.

***** Reviewer's comments *****

Referee #1 (Remarks):

The manuscript by Harhour et al. presents new findings related to the Hutchinson-Gilford Progeria Syndrome, a rare premature aging disease. The authors demonstrate that progerin, the disease causing protein, is sequestered into abnormally shaped PML-NBs. Since PML-NBs have been implicated in protein degradation via the UPS, the authors reasoned that progerin might be degraded in those compartments via UPS. To test this hypothesis, the authors treated HGPS fibroblasts with proteasome inhibitors for 24h and monitored progerin levels in the cells. Interestingly, MG132 treatment resulted in the decrease of the progerin levels, partially through induction of progerin degradation by macroautophagy and by reducing progerin mRNA expression through reduction of SRSF-1 levels. The authors go on to demonstrate that MG132 injection in the skeletal muscle of LmnaG609G/G609G mice locally reduces SRSF-1 expression and progerin levels.

Overall, the manuscript contains interesting findings for the HGPS field, however, there are two major issues, one relates to the science and the other to the organization of the manuscript.

Science:

From the presented data, it is not clear what the mechanisms of the MRG132 effect are. The authors emphasize a pathway through the splicing factor SRSF1; however, several pieces of data point to a direct protein degradation effect and other data point to a yet to be identified pathway. There is considerable uncertainty as to what the mechanistic basis of the observed effects is.

For example, in figure 1C the authors show reduction of 31%, 58% and 65% of progerin at 6, 24, 48h. Given a half-life of more than 24h for progerin, these are clearly not SRSF1-mediated effects, but are due to protein turnover. While macroautophagy may account for this, it is not clear from the presented data that it does. Furthermore, how MG132 promotes this dramatic macroautophagy effect is not clear.

I was puzzled by the statement that cytoplasmic transport was not required for the MG132-mediated clearance of progerin. This would then mean that the protein is degraded in the nucleus, which would rule out macroautophagy. What is this additional nuclear degradation pathway?

Furthermore, Fig. 4B shows approximately 90% downregulation of the progerin mRNA levels at 24h when already 50% of protein reduction is observed in Fig. 4C. Since progerin has a long half-life, the observed decrease in protein levels already at 24h suggests that another pathway is involved in progerin clearance prior to the SRSF-1-induced transcriptional inhibition. Macroautophagy could partially explain progerin protein clearance at 24h, but since MG132 treatment in combination with autophagy inhibitors did not completely restore progerin protein levels in Fig. 4F, there is a strong possibility of a third pathway involved in progerin clearance. One possibility are caspases. Since pan-caspase inhibitors alone can completely restore progerin protein levels, is it possible that caspases are involved in progerin clearance in addition to the macroautophagy and SRSF1. It would be important to identify additional contributors to progerin reduction.

The authors suggest that SRSF1 is involved in the use of the cryptic splice site that creates progerin RNA. One prediction from this model would be that elimination of SRSF1 should lead to the production of more wild-type lamin A RNA since the cryptic splice site is not used anymore. This is not observed. The authors also observe reduction of lamin A/C RNA and it is not clear why that would be.

In the Fig. 4C, the authors observe normalization of progerin and SRSF-1 levels 72h and 96h post-treatment. What happens if cells are treated with MG132 for longer time points (e.g. 48h, 72h), remain progerin and SRSF-1 levels down?

The data on the rescue of HGPS disease phenotypes upon MG132 treatment need more work. The proliferation rescue is not extensive. What is the extent of the proliferation phenotype compared to age-matched WT control? I would also test the rescue of additional cellular phenotypes (e.g. LAP2, LB1, HP1 α ...). In addition, I would replace "disease phenotypes" in the text with "cellular HGPS phenotypes".

It is premature to propose MG132 for therapeutic purposes. As pointed out above, the mechanism involved is far from clear, MG132 is unlikely to be without severe side effects and, most importantly, additional *in vivo* data on the effect of MG132 on ameliorating HGPS phenotypes (longevity, vasculature, bone etc.) are required for this discussion.

Organization:

The manuscript contains large amounts of various data put together, but the presentation seems arbitrary, lacking flow with the authors jumping from one part to another without a clear connection. Therefore, I would suggest to focus only on the MG132 aspect of the data and remove the PML-NBs and sumoylation data which could be used for a separate publication.

Referee #2 (Comments on Novelty/Model System):

I enjoyed reading this study, and consider it novel enough for publication, if the authors can address issues with the results and then tailor changes to the discussion appropriately after that is performed.

Referee #2 (Remarks):

Overall, I find this investigation fascinating. Researchers in the Progeria field have tended to ignore the speckles seen in the nucleoplasm, and I have always wondered if these instead were a significant component of disease in Progeria. I applaud the authors for not only identifying the progerin component of these speckles, but also delving deeply into the mechanisms by which these operate and can potentially be manipulated to the benefit of the cells (and potentially the patients). I also find this study well-written overall. However, I have a number of issues with the data, all of which I hope can be addressed both in the results section and in the discussion.

Brief overview of statements and claims:

The first claim in this study is that progerin is sequestered into PML-NBs (intranuclear speckles that are associated with the nuclear matrix, including lamin A), and that these can serve as novel biomarkers for HGPS.

The study then claims it demonstrates that MG132, a general proteasome inhibitor, has effects on progerin via two distinct mechanisms, decreased production and increased degradation.

1. MG132 causes progerin degradation via macroautophagy
2. MG132 downregulates SRSF-1, which causes a decrease in abnormal prelamin A mRNA splicing for progerin, yielding decreased progerin production

The claims also include downstream improvements in disease phenotype include decreased senescence and increased proliferation in cultured fibroblasts.

Further, the study claims that IM injection of MG132 into a murine model of HGPS yielded decreased SRF-1 expression and decreased progerin tissue content.

Issues to address:

Results section

Figure EV1A - Was the control line staining negative, supporting this staining shown is progerin?

Figure EV1B - PMN counting morphology data is unclear to me. Are these means of subjectively counted numbers of punctate structures per cell? I don't see a protocol for this. Also, the graph seems to be based on a single cell line, single count, as I don't see any error bars. Is this the case? If so I suggest you conduct repeat n's to support your singular but very important observation.

You claim this ring and thread-like morphology is a novel biomarker in HGPS. This may be true and important. Please clarify in your text whether this observation is novel to all disease/non-diseased cells. In other words, is this morphological change seen anywhere else?

Figure EV1C - do normal fibroblasts have this same staining or different staining? Please show

Figure EV1E - It is unclear if you did you stain for calreticulin. Please clarify and give ns.

Figure EV2 - it's difficult to see lamin A in the nucleus here, where it should be present even if it is not located within the PML-NBs. Please explain or show a better photo. Also, please include a control line with similar staining. Is this colocalization present in control lines as well?

Figure 1 shows progerin staining on Progeria fibroblasts. Inhibition of proteasome activity paradoxically decreased progerin content, and cell viability was not responsible.

1A Please include a control panel or statement that you tested this on control cells and got no staining, thus supporting that this antibody is progerin-specific.

Figure EV4. Please confirm that you performed these test on control cells and that progerin was not detected.

Bortezomib and Xarfilzomib, 2 drugs that inhibit proteasome activity through the ubiquitin pathway, did not effect progerin levels in vitro.

Figure EV5 dissects the influences of sumoylation and ubiquitination on progerin susceptibility to proteasome inhibitors. Only sumoylated and not sumoylated plus ubiquitinated progerin was susceptible to MG132; thus explaining the failure of Bortezomib and Xarfilzomib.

Figure 1D shows that inhibiting autophagy with chloroquine blocks the MG132 activity on progerin. I could not see the LC3B-I vs. II bands well. Can you provide a graph showing ratios?

"The altered cellular phenotype that characterizes HGPS patients' cells, going along with progerin accumulation, and their premature senescence in culture, led us to investigate the subcellular localization of progerin during prolonged fibroblast cell passages."

Please give a reference that shows actual progerin accumulation simultaneous with cellular phenotype abnormality and premature senescence.

In section lines 209-214, and Figure 4A you state that chloroquine does not have an effect on MG132 mediated progerin clearance, but in Figure 1D you show it does. Please clarify.

Figure 4C - your graph shows that progerin decreases at 6 hours prior to SRSF-1 decrease. This does not support your hypothesis since SRSF-1 should decrease first. Please explain or change your explanation of the events.

Figure 4E - why does your control siRNA reduce progerin and SRSF-1? Your control shows nonspecific reduction and therefore your siRNA experiment does not look valid to me.

The suppression of SRFS-1 is expected to decrease progerin splicing in favor of lamin A splicing, thus decreasing progerin and increasing lamin A protein.

Figure EV6 shows that treatment with MG132, which was hypothesized to suppress SRFS-1, lead to decreased lamin A splicing instead of the expected increase. Protein levels did not follow this pattern, but he increase in lamin A protein was quite mild. Thus, the splicing mechanism is as yet unsupported, and the protein changes cannot be explained via splicing. This is not stated and this data is in the manuscript but no good explanation is offered.

Between figures 4C,E and EV6 , I do not see good support for your splicing hypothesis. Please address.

Figures 4A and F - what constitutes a + vs - scoring? What is the threshold? How many times did you perform this experiment?

Figure 4A and F - in the graph, what are the X-axis points? Time points, different treatments?
Figure 5 - there are automated methods for scoring nuclei now. You used an older method of "eyeballing" the nuclei. Can you automate this?

Figure 5D - did you show that the untreated or DMSO-treated cultures increased senescence over the 10 days? I don't see this in the graph. Yet you state in the text that you "rescued" senescence. This does not support a rescue of senescence but perhaps a decrease with treatment in the number of senescing cells. In fact, day 10 culture has about the same amount of senescence in the treated culture as day 1. Please respond.

Figure 5F - I assume the * indicates comparison with DMSO bar. However, that bar starts out lower on day 1 than MG132. Therefore, I'm concerned that you are not accurate when you state that the relative difference on day 10 indicates improved viability with MG132. The same holds for 5E. You need to show that these cultures start equally, pre-treatment and you have not done that.

Fig 6A - again you need to state or show staining in control line to prove progerin specificity of your antibody.

Figure 7 - please make the Y axes for B and C similar. It is hard to compare these two doses otherwise. Why are your untreated bars so different from B to C? Shouldn't they be similar, as they come from untreated muscle in mice with the same mutation? Please address this important issue.

As noted, lamin A is not affected significantly, though the prior description of splicing effect would lead us to hypothesize that lamin A splicing would increase.

Progerin is suppressed much more in C at the higher dose, but SRSF-1 is not. SRSF-1 suppression is similar between B and C. I would expect an increased SRSF-1 suppression to move along with increased progerin splicing suppression. This argues against the influence of SRSF-1.

Discussion:

Once you address the issues with data and data presentation, please review the discussion points to assure they accurately reflect the results. Most particularly, your discussion points about the SRSF-1 splicing effect, but also the other issues raised above. You should clarify your data on shape, senescence, viability and proliferation, before stating that these significantly improved with MG132 treatment, or soften that statement in the discussion.

You state that progerin cannot be degraded when it is sumoylated. I think this is an overstatement. Please soften to state that it is not degraded by similar means when it is sumoylated vs when it is not.

You state the MG132 can have benefit without toxicity in vivo. Do you refer only to the references in the paragraph to make this statement? Please clarify the statement.

The paragraph on IM MG132 treatment does not address the issues raised above regarding results. Please do so. Also, I don't see how lamin A "compensates" for lamin C reduction. This global statement should be omitted.

References:

There is a more recent and larger study of mean age of death in progeria which calculates 14.6 years (Gordon et al, 2012). Please update this information appropriately.

Primary fibroblasts from HGPS patients exhibit reduced proliferation as well as premature senescence (Huang et al, 2005). This paper does not show these properties in relation to normal fibroblasts. In fact, only HGPS cells are used in this paper. Please improve your reference.

I didn't perform an exhaustive reference check, so I urge the authors to please do so since I did find two mistakes fairly easily.

Referee #3 (Remarks):

The manuscript by Harhour and co-workers shows that the proteasome inhibitor MG132 - and some related compounds - surprisingly reduces the levels of progerin in cells from HGPS patients. The proposed mechanism is that MG132 induces autophagy which leads to progerin degradation and that it simultaneously reduces progerin production by interfering with SRSF-1 - a protein that stimulates progerin splicing from the prelamin A message. MG132 thereby improved some of the cellular phenotypes of HGPS, such as the impaired proliferation and reduced viability. Overall, the experiments are well designed and the results provide new insight into the biology of progerin handling in cells and indirectly into the pathogenesis of HGPS. This paper continues a long line of novel discoveries from this leading group of investigators in the field.

There are some concerns. One concern - which is difficult to address at this point -- is that the paper contains several different findings, several of which are left unexplained and others which are not coupled to the main story. Therefore, the paper appears a bit unfocused. Other than that, there are a few points that can be addressed:

Major:

1. A puzzling result in this paper is that many of the figures (e.g., 1A, C, and D; 4A and F; 6A and B etc.) demonstrate that MG132 dramatically reduces progerin levels; in a substantial proportion of western blots, progerin is essentially gone. Regardless, the effect of MG132 treatment on all of the well-known cellular phenotypes of HGPS cells is modest (e.g., Fig 5). Reducing the levels of a toxic lamin A mutant (progerin in some studies, prelamin A in others) produces much more dramatic effects in earlier studies. So, why didn't the substantial reduction in progerin by MG132 result in more robust cellular effects? Could beneficial progerin-reducing MG132 effects on HGPS cells be masked by other deleterious drug-effects? This needs to be addressed with both experiments and discussion. For example, compare the effect of MG132 on cell phenotypes with cells where progerin expression is reduced with si- or sh-RNAs targeting progerin/prelamin A? (to the same levels as in MG132-treated cells). And, would MG132 treatment reduce the levels of full-length prelamin A that accumulate in patients with related progeroid disorders (where splicing is not involved)?
2. In Fig. 1D; did the chloroquine treatment result in significantly higher levels of progerin compared to MG132 alone or do the stars only concern the comparison with the high levels of progerin in untreated HGPS cells?
3. Lines 352-356: The mechanisms underlying MG132 effects on progerin levels needs to be addressed and discussed better because the 5.5-page discussion doesn't do the job of succinctly describing it. You should use the Fig EV7 more effectively in combination with the text to accomplish this.

If the effect is unrelated to proteasome inhibition, by what mechanisms would MG132, but not BTZ or CFZ, stimulate macroautophagy or nuclear-to-cytoplasm shuttling? Because the chloroquine effect was modest, is the MG132-effect mainly mediated by the reduced SRSF-1 levels and the reduced progerin mRNA levels?

4. Does the MG132 effect require progerin to be present in abnormal PMLs? Would MG132 treatment reduce progerin levels in early passage cells, before the appearance of abnormal PMLs?
5. Data in Fig. 5E and F: With the normalization at each time point, it is difficult to judge the potential cumulative effect of the treatment over time. Conventional cell proliferation assays should be shown here. This is important as you conclude that MG132 significantly enhanced proliferation and reduced senescence of HGPS cell lines.
6. Lines 7, 93, and 335: A biomarker that does not appear in the cells until after 12-20 passages is not particularly useful. The late appearance of the abnormal PML-NBs should be commented on when mentioning biomarkers. Their late appearance in cell-culture dishes also suggest the possibility that they are not relevant in vivo. However, they are clearly specific to the HGPS cells and the careful analyses of their contents and dynamics allowed the investigators to uncover important new biology.

7. Many of the nuclei shown in regular and expanded view figures show only minor shape

abnormalities. Can you comment on whether the findings on progerin localization/co-localization hold up in grossly abnormally-shaped nuclei and is there a correlation between abnormal PMLs and blebbing?

Minor comments:

1. Fix lots of spelling mistakes: e.g., line 42 "heterodimerize"; line 954 "de"; line 344 "benn"; EVF 1E, "Transpatent"
2. The legend to Figure EV5 B and C do not match the labeling in the figure (they have been switched).
3. Lines 954, 955: was the medium replaced with new drug every 48 hours? If so, change the text to make that clear.

1st Revision - authors' response

21 April 2017

Referee #1 (Remarks):

The manuscript by Harhour et al. presents new findings related to the Hutchinson-Gilford Progeria Syndrome, a rare premature aging disease. The authors demonstrate that progerin, the disease causing protein, is sequestered into abnormally shaped PML-NBs. Since PML-NBs have been implicated in protein degradation via the UPS, the authors reasoned that progerin might be degraded in those compartments via UPS. To test this hypothesis, the authors treated HGPS fibroblasts with proteasome inhibitors for 24h and monitored progerin levels in the cells. Interestingly, MG132 treatment resulted in the decrease of the progerin levels, partially through induction of progerin degradation by macroautophagy and by reducing progerin mRNA expression through reduction of SRSF-1 levels. The authors go on to demonstrate that MG132 injection in the skeletal muscle of LmnaG609G/G609G mice locally reduces SRSF-1 expression and progerin levels.

Overall, the manuscript contains interesting findings for the HGPS field, however, there are two major issues, one relates to the science and the other to the organization of the manuscript.

We thank this referee for his/her comments with constructive suggestions and now address all points in the revised manuscript.

Science:

From the presented data, it is not clear what the mechanisms of the MRG132 effect are. The authors emphasize a pathway through the splicing factor SRSF1; however, several pieces of data point to a direct protein degradation effect and other data point to a yet to be identified pathway. There is considerable uncertainty as to what the mechanistic basis of the observed effects is.

For example, in figure 1C the authors show reduction of 31%, 58% and 65% of progerin at 6, 24, 48h. Given a half-life of more than 24h for progerin, these are clearly not SRSF1-mediated effects, but are due to protein turnover. While macroautophagy may account for this, it is not clear from the presented data that it does.

We agree with the referee that different mechanisms involved in MG132-mediated progerin clearance (autophagy, caspases, splicing) were approached separately at the beginning of the manuscript, to finally show that the blockade of both autophagy and caspase-mediated cleavage of the progerin splicing regulator SRSF-1, allowed the complete restoration of basal progerin levels, confirming the dual action of MG132 leading to progerin clearance. What is not clear enough could be the reduction in progerin levels (mRNA and protein) at 6 h after treatment in the absence of SRSF-1 levels decrease. In order to better dissect the mechanisms involved during the first hours of treatment, we carried out complementary experiments (new Figure 4E) and showed that another splice factor (SRSF-5), inhibiting progerin production (Vautrot et al, 2016) (unlike SRSF-1), is increased from 4 h of treatment and this increase correlates with the decrease of transcripts and consequently of progerin proteins levels from 6 h post-treatment.

In summary, in primary cultures of fibroblasts from HGPS patients, progerin is sequestered in PML-NBs. Treatment of these cells with MG132 initially induces a caspase-dependent reduction of SRSF-1 levels as well as a rapid but not sustained increase of SRSF-5 levels even after the treatment renewal. The modulation of the levels of these two splicing factors is in favor of a decrease in progerin levels. In parallel, MG132 at first induces nucleolar translocation of progerin and then its accumulation and degradation in autophagic vacuoles.

Measured half-lives of specific proteins should not be considered as absolute measurements, but rather as an estimation of the stability of characterized proteins. Such predictions raise questions of what extent reported protein half-lives can be extrapolated to other experimental systems. Indeed, there is a turnover rate diversity between different experiments involving a high degree of regulation in which protein stabilities are modified. In fact, the half-life of progerin described in Youn-Sang Jung paper (approximately 20h) is largely decreased to 3.5 hours following treatment with the von Hippel-Lindau tumor suppressor also known as pVHL (Jung et al, 2013), or by a compound that block progerin-lamin A/C binding (Lee et al, 2016). These observations are consistent with a possible change in progerin half-life in response to MG132 treatment. Moreover, as shown in Fig. 1C, treatment with DMSO does not induce any decrease in progerin levels under control conditions. Thus, eliminating a possible involvement of progerin turnover in its clearance. Moreover, MG132 acts not only at the progerin synthesis by modulating SRSF-1 and SRSF-5, but also at its degradation. It is therefore legitimate that this treatment modifies progerin half-life.

Furthermore, how MG132 promotes this dramatic macroautophagy effect is not clear.

We thank this referee for bringing up this important question. It is known that proteasome inhibitors induce autophagy as a compensatory response (Tang et al, 2014; Zang et al, 2012; Zhu et al, 2010). Several potential explanations have been suggested to explain the exact mechanism(s) of this cross-talk. One such proposed mechanism involving induction of the unfolded protein response (UPR) or inhibition of mTOR and induction of damage-regulated autophagy modifier (Korolchuk et al, 2010). Further, we showed in our manuscript that autophagy activation is supported by the increased amounts of LC3B-II/LC3BI ratios on immunoblotting assays, progerin delocalization into cytoplasmic autophagic vacuoles, increased autophagic transcript levels using RNA seq experiments and partial restoration of progerin levels in presence of Chloroquine or Bafilomycin A1 which are commonly used as autophagy inhibitors.

I was puzzled by the statement that cytoplasmic transport was not required for the MG132-mediated clearance of progerin. This would then mean that the protein is degraded in the nucleus, which would rule out macroautophagy. What is this additional nuclear degradation pathway?

Using leptomycin B (an inhibitor of the exportin-1 dependent nuclear export through the **nuclear pores** (Fukuda et al, 1997)), we showed that cytoplasmic transport through the nuclear pores was not required for the MG132-mediated clearance of progerin. However, emerging data suggest that others mechanisms deliver nuclear targets at the nuclear envelope to the cytoplasmic autophagy pathways. Indeed, it has been shown that lamin B1 degradation is also achieved by nucleus-to-cytoplasm transport that delivers lamin B1 to the lysosome without passing through the nuclear pores, meaning that nuclear materials to be degraded can be encapsulated through a process similar to exocytosis, and this process has been widely described (Boban & Foisner, 2016; Dou et al, 2015; King & Lusk, 2016; Luo et al, 2016).

Furthermore, Fig. 4B shows approximately 90% downregulation of the progerin mRNA levels at 24h when already 50% of protein reduction is observed in Fig. 4C. Since progerin has a long half-life, the observed decrease in protein levels already at 24h suggests that another pathway is involved in progerin clearance prior to the SRSF-1-induced transcriptional inhibition. Macroautophagy could partially explain progerin protein clearance at 24h, but since MG132 treatment in combination with autophagy inhibitors did not completely restore progerin protein levels in Fig. 4F, there is a strong possibility of a third pathway involved in progerin clearance. One possibility are caspases. Since pan-caspase inhibitors alone can completely restore progerin protein levels, is it possible that caspases are involved in progerin clearance in addition to the macroautophagy and SRSF1. It would be important to identify additional contributors to progerin reduction.

We agree with the referee that macroautophagy could partially explain progerin protein clearance at 24h, since LC3BII/LC3BI ratio is increased starting after 2 hours treatment (New Fig. 4E). As we have answered above, we further investigated the mechanisms involved in MG132-mediated progerin clearance and we found that SRSF-5 accumulation from 4 h of MG132 treatment could potentially explain the rapid decrease in progerin transcripts and protein levels. Consistent with data already published, in cells exposed to pharmacological treatments, the progerin half-life is likely to vary, or even to decrease to 3.5h instead of 20h. Therefore, the decrease in SRSF-1 levels observed at 24h and the increase in SRSF-5 levels from 4 hours after treatment, in addition to the increase in the LC3BI / LC3BII ratio appear to be compatible with progerin clearance before 24h. Concerning Caspases involvement in progerin clearance, Caspase-6 is widely recognized as being responsible of lamins A/C cleavage during apoptosis. Lamin A/C as well as progerin exhibit a conserved VEID caspase 6 cleavage site (Slee et al, 2001). To investigate the involvement of caspase-6 in MG132-induced progerin clearance, we treated HGPS cells for 48h with a combination of MG132 and a caspase-6 inhibitor. Again, progerin levels remained below the HGPS control even in the presence of autophagy inhibition by chloroquine (Fig 4A, Well⁹) suggesting that caspase-6 is not involved in progerin clearance. As indicated in Fig. 4G, Well⁸, pan-caspase inhibitors alone (without autophagy inhibitor) can't completely restore progerin protein levels. Progerin levels are completely restored only after exposure of the cells to both the autophagic and pan-caspase (in particular caspase 3 responsible for cleavage of SRSF-1) inhibitors (Fig. 4G, Well⁹).

The authors suggest that SRSF1 is involved in the use of the cryptic splice site that creates progerin RNA. One prediction from this model would be that elimination of SRSF1 should lead to the production of more wild-type lamin A RNA since the cryptic splice site is not used anymore. This is not observed. The authors also observe reduction of lamin A/C RNA and it is not clear why that would be.

The reduction of lamin A transcripts is not surprising given that the lamin A 5'SS is also recognized by SRSF-1 independently of the HGPS mutation that promotes the reinforcement of another weak 5' splice-site also recognized by SRSF-1 and leading to progerin production. SRSF1 elimination has already been described to decrease in progerin, lamin A and lamin C transcripts levels in HGPS cells upon treatment with Metformin (Egesipe et al, 2016).

In the Fig. 4C, the authors observe normalization of progerin and SRSF-1 levels 72h and 96h post-treatment. What happens if cells are treated with MG132 for longer time points (e.g. 48h, 72h), remain progerin and SRSF-1 levels down?

Thank you for this relevant suggestion. As requested, we now provide additional experiment on the effect of MG132 treatment renewal on progerin and SRSF-1 levels. By renewing treatment after 48h, we show that progerin and SRSF-1 levels remain down (Fig 4E).

The data on the rescue of HGPS disease phenotypes upon MG132 treatment need more work. The proliferation rescue is not extensive. What is the extent of the proliferation phenotype compared to age-matched WT control? I would also test the rescue of additional cellular phenotypes (e.g. LAP2, LB1, HP1 α ...). In addition, I would replace "disease phenotypes" in the text with "cellular HGPS phenotypes". It is premature to propose MG132 for therapeutic purposes. As pointed out above, the mechanism involved is far from clear, MG132 is unlikely to be without severe side effects and, most importantly, additional in vivo data on the effect of MG132 on ameliorating HGPS phenotypes (longevity, vasculature, bone etc.) are required for this discussion.

We thank this referee for this justified criticism and provide additional experiments on this issue. Using a conventional cell proliferation assay (BrdU), we show that MG132 treatment scheme increased proliferation rate of HGPS cells with a curve approximately similar, but without reaching the proliferation rate of age-matched WT control cells (New Fig 5E). As requested, we investigated the rescue of additional HGPS cellular phenotypes (γ -H2AX, Tri-Me-K9, LAP2 α , lamin B1 and HP1 α). As presented in the new Fig.5H, treatment of HGPS cells with MG132 improved these cellular phenotypes with a rescue level comparable to that of age-matched WT control cells or HGPS cell where progerin expression is reduced with antisense oligonucleotides (AON) targeting progerin. Furthermore, we performed an RNA-seq experiment and analyzed the expression levels of genes that are misregulated in HGPS fibroblasts. Interestingly, we found an increase in the transcripts levels of genes involved in autophagy activation (p62, ATG4A, ATG4B, ATG4D,

ATG3, ATG14, LLK1), proteasome subunits production (PSMC4, PSMB3), splicing (SRSF5), metalloprotease transcripts including MMP3 (whose levels are decreased in HGPS cells), growth factors (EGF, FGF17, FGF22), Apoptosis inhibitors (BAG3, BFAR) as well as the improvement of other transcripts involved in inflammation (IkB, SIRT6) (New Fig 5I).

As requested, we replaced "disease phenotypes" with "cellular HGPS phenotypes".

We agree that additional *in vivo* data on the effect of MG132 on ameliorating HGPS phenotypes (longevity, vasculature, bone etc.) are required before proposing MG132 or its analogs for therapeutic purposes. Systemic injections of MG132 in *Lmna*^{G609G/G609G} mice, which of course will be needed to target the different organs and characteristics involved in the pathophysiology of progeria, will likely require the setting up of an appropriate galenic form of this short peptide in order to increase its half-life. We are currently working with chemists on the optimization of these tri-peptides (MG132 and its analogs) in order to perform long-term systemic studies. Considering the amount of work, these additional *in vivo* studies could be the subject of another article.

Organization:

The manuscript contains large amounts of various data put together, but the presentation seems arbitrary, lacking flow with the authors jumping from one part to another without a clear connection. Therefore, I would suggest to focus only on the MG132 aspect of the data and remove the PML-NBs and sumoylation data which could be used for a separate publication.

We agree with the referee that several degradation pathways were addressed in the manuscript, but the final goal was to determine, within a complete story, the mechanisms involved in the accumulation and clearance of progerin. We also agree that the PML-NBs and sumoylation data could be used for a separate publication but we think that it is necessary to keep the exploration of the PML-NBs since they are involved in progerin sequestration and that the treatment that we used (MG132) induces, among others, the delocalization of progerin from these PML-NBs and its clearance.

Referee #2 (Comments on Novelty/Model System):

I enjoyed reading this study, and consider it novel enough for publication, if the authors can address issues with the results and then tailor changes to the discussion appropriately after that is performed.

Referee #2 (Remarks):

Overall, I find this investigation fascinating. Researchers in the Progeria field have tended to ignore the speckles seen in the nucleoplasm, and I have always wondered if these instead were a significant component of disease in Progeria. I applaud the authors for not only identifying the progerin component of these speckles, but also delving deeply into the mechanisms by which these operate and can potentially be manipulated to the benefit of the cells (and potentially the patients). I also find this study well-written overall. However, I have a number of issues with the data, all of which I hope can be addressed both in the results section and in the discussion.

Brief overview of statements and claims:

The first claim in this study is that progerin is sequestered into PML-NBs (intranuclear speckles that are associated with the nuclear matrix, including lamin A), and that these can serve as novel biomarkers for HGPS. The study then claims it demonstrates that MG132, a general proteasome inhibitor, has effects on progerin via two distinct mechanisms, decreased production and increased degradation.

1. MG132 causes progerin degradation via macroautophagy
2. MG132 downregulates SRSF-1, which causes a decrease in abnormal prelamin A mRNA splicing for progerin, yielding decreased progerin production

The claims also include downstream improvements in disease phenotype include decreased senescence and increased proliferation in cultured fibroblasts.

Further, the study claims that IM injection of MG132 into a murine model of HGPS yielded decreased SRF-1 expression and decreased progerin tissue content.

We thank this referee for his/her highly encouraging and positive remarks with constructive suggestions and greatly appreciate his/her interest in our study.

Issues to address:

Results section

Figure EV1A - Was the control line staining negative, supporting this staining shown is progerin?

Thank you for expressing this concern. As requested, we provide new images (Fig. EV1A) showing negative progerin staining on control cells line from a healthy individual.

Figure EV1B - PMN counting morphology data is unclear to me. Are these means of subjectively counted numbers of punctate structures per cell? I don't see a protocol for this. Also, the graph seems to be based on a single cell line, single count, as I don't see any error bars. Is this the case? If so I suggest you conduct repeat n's to support your singular but very important observation.

We agree that this data was missing and should be added. The percentage of abnormal PML-NBs (ring-like and thread-like) in fibroblasts from four HGPS patients and four normal individuals matched for age and passage, was calculated by two independent observers using a manual blind counting. At least 200 fibroblast nuclei were randomly selected for each cell line (n=4) and examined. Results are now expressed graphically as mean \pm SEM of the average percentage of the total nuclei counted.

You claim this ring and thread-like morphology is a novel biomarker in HGPS. This may be true and important. Please clarify in your text whether this observation is novel to all disease/non-diseased cells. In other words, is this morphological change seen anywhere else?

We thank the referee for pointing this out. Structural changes within PML-NBs have been shown to occur during DNA and RNA viral infections (reviewed in (Doucas & Evans, 1996); (Sternsdorf et al, 1997)) since PML-NBs are sites for the early stages of transcription and replication of DNA and RNA viruses and are also sites for the subsequent cellular antiviral defense mechanisms using IFNs. Dramatic linear and rosette PML-NBs lacking substantial SUMO-1, Daxx, and Sp100 have been described as unique to early hESC cultures. These occur primarily between Day 0–2 of differentiation and become rare thereafter (Butler et al, 2009). PML bodies are larger in A-type lamin-deficient fibroblasts compared with their WT counterparts (Stixova et al, 2012).

This part was included in the discussion.

Figure EV1C - do normal fibroblasts have this same staining or different staining? Please show

As requested, we provide new images (Fig. EV1C) showing negative progerin staining and normal PML staining on control cells line from a healthy individual.

Figure EV1E - It is unclear if you did you stain for calreticulin. Please clarify and give ns.

As requested, we performed PML and calreticulin staining and show that calreticulin is not included within PML-NBs in HGPS fibroblasts nor in control cell lines (Fig EV1E, lower Panel).

Figure EV2 - it's difficult to see lamin A in the nucleus here, where it should be present even if it is not located within the PML-NBs. Please explain or show a better photo. Also, please include a control line with similar staining. Is this colocalization present in control lines as well?

We replaced lamin A staining with a better photo, illustrating lamin A localization in the nucleus. As requested, we now provide a control line stained with the same antibodies (Lamin B1/B2, Emerin, Lamin A, Lamin C and Nup-153) as in HGPS fibroblasts (Fig. EV2A). There are no PML-NBs labeling with these antibodies in control lines.

Figure 1 shows progerin staining on Progeria fibroblasts. Inhibition of proteasome activity paradoxically decreased progerin content, and cell viability was not responsible.

1A Please include a control panel or statement that you tested this on control cells and got no staining, thus supporting that this antibody is progerin-specific.

Thank you for expressing this concern. As requested, we provide new images (Fig. EV1A) showing negative progerin staining on control cells line from a healthy individual and supporting the specificity of progerin antibody.

Figure EV4. Please confirm that you performed these test on control cells and that progerin was not detected.

Progerin specificity on Western-blot experiments has been validated in Fig.1C, using control cells line from a healthy individual.

Bortezomib and Xarfilzomib, 2 drugs that inhibit proteasome activity through the ubiquitin pathway, did not effect progerin levels in vitro.

Figure EV5 dissects the influences of sumoylation and ubiquitination on progerin susceptibility to proteasome inhibitors. Only sumoylated and not sumoylated plus ubiquitinated progerin was susceptible to MG132; thus explaining the failure of Bortezomib and Xarfilzomib.

We thank the referee for these relevant suggestions. We would like to clarify that MG132 elicited progerin decrease only when this was ubiquitinated, but not when it was both sumoylated and ubiquitinated (Fig.EV5 B and C). To better determine whether progerin sumoylation is required for its clearance, we treated HGPS fibroblasts with sumoylation inhibitors (either 2D08 or ginkgolic acid) in combination with proteasome inhibitors (MG132, MG262, MG115, Bortezomib or Carfilzomib). The results presented in Fig. EV5E show that progerin levels are not significantly affected in the presence or absence of sumoylation inhibitors in response to treatment with MG132, MG115 or Bortezomib. Under the same conditions, the effect of MG262 on progerin clearance is more pronounced when sumoylation is inhibited. On the contrary, sumoylation of progerin appears to be required to achieve its clearance in response to Carfilzomib.

Figure 1D shows that inhibiting autophagy with chloroquine blocks the MG132 activity on progerin. I could not see the LC3B-I vs. II bands well. Can you provide a graph showing ratios?

We agree that this data was missing and should be added. As requested, we now provide a graph showing LC3BII/LC3BI ratios (Fig. 1D).

“The altered cellular phenotype that characterizes HGPS patients' cells, going along with progerin accumulation, and their premature senescence in culture, led us to investigate the subcellular localization of progerin during prolonged fibroblast cell passages.”

Please give a reference that shows actual progerin accumulation simultaneous with cellular phenotype abnormality and premature senescence.

We thank the reviewer for pointing this out. The cause-and-effect relationship between progerin accumulation and senescence has already been determined (Goldman et al, 2004; Scaffidi & Misteli, 2005; Scaffidi & Misteli, 2008; Vidak & Foisner, 2016). As requested, these references were added to the revised manuscript.

In section lines 209-214, and Figure 4A you state that chloroquine does not have an effect on MG132 mediated progerin clearance, but in Figure 1D you show it does. Please clarify.

We apologize for the confusion. To make reading through this section clearer, we replaced it with: “To investigate the involvement of caspase-6 in MG132-induced progerin clearance, we treated HGPS cells for 48h with a combination of MG132 and a caspase-6 inhibitor. Again, progerin levels remained below the HGPS control even in the presence of autophagy inhibition by chloroquine (Fig

4A, Well⁹), suggesting that another pathway, in addition to autophagy, was involved in progerin levels' reduction upon MG132".

Figure 4C - your graph shows that progerin decreases at 6 hours prior to SRSF-1 decrease. This does not support your hypothesis since SRSF-1 should decrease first. Please explain or change your explanation of the events.

We agree that the reduction in progerin levels (mRNA and protein) at 6 h after treatment cannot be correlated to SRSF-1 levels whose decrease was observed at 24h. In order to better dissect the mechanisms involved during the first hours of treatment, we carried out complementary experiments (Figure 4E) and showed that another splice factor (SRSF-5), inhibiting progerin production (Vautrot et al, 2016) (unlike SRSF-1), is increased from 4 h of treatment and this increase correlates with the decrease of transcripts and consequently of progerin proteins levels from 6 h post-treatment.

Figure 4E - why does your control siRNA reduce progerin and SRSF-1? Your control shows nonspecific reduction and therefore your siRNA experiment does not look valid to me.

We are sorry for the lack of clarity of this figure legend (which now corresponds to Fig. 4F). We now explain better the experimental protocol by indicating that cells transfected with siRNA control were also treated with either DMSO (indicated by the - sign) or MG132 (indicated by the + sign). In this last condition (siRNA Control + MG132 treatment), the decrease in SRSF-1 is due to MG132 but not to siRNA control.

The suppression of SRSF-1 is expected to decrease progerin splicing in favor of lamin A splicing, thus decreasing progerin and increasing lamin A protein.

Figure EV6 shows that treatment with MG132, which was hypothesized to suppress SRSF-1, lead to decreased lamin A splicing instead of the expected increase. Protein levels did not follow this pattern, but the increase in lamin A protein was quite mild. Thus, the splicing mechanism is as yet unsupported, and the protein changes cannot be explained via splicing. This is not stated and this data is in the manuscript but no good explanation is offered.

The reduction of lamin A transcripts is not surprising given that the lamin A 5'SS is also recognized by SRSF-1 independently of the HGPS mutation that promotes the reinforcement of another weak 5' splice-site also recognized by SRSF-1 and leading to progerin production. SRSF1 elimination has already been described to have an effect on the decrease in progerin, lamin A and lamin C transcripts levels in HGPS cells (Egesipe et al, 2016).

Between figures 4C,E and EV6 , I do not see good support for your splicing hypothesis. Please address.

In Fig. 4C, we show that following MG132 treatment, SRSF-1 levels are decreased and this decrease is concomitant with progerin levels decrease. When the molecule loses its activity after 72 hours and 96 hours of treatment (evidenced by the re-increase in the proteasome activities starting at 72h, Fig. 4D), SRSF-1 levels increase is also concomitant with progerin levels increase.

When the treatment is renewed after 48 h, SRSF-1 levels decrease and consequently progerin levels decrease are maintained (new Fig. 4E).

In Fig. 4F (formerly Fig. 4E), we show that SRSF-1 siRNA results in decreased progerin levels and that the combination of SRSF-1 siRNA with MG132 treatment has an additive effect inducing a greater decrease of progerin levels.

These observations are in favor of a role SRSF-1 reduction after MG132 treatment in decreasing progerin levels.

Referring to Fig. EV6, we show that MG132 treatment also induces a decrease in lamin A and lamin C transcripts levels (already described following Metformin-mediated SRSF-1 levels decrease) (Egesipe et al, 2016) and that these decreases in transcripts aren't correlated with protein levels. The poor correlation between transcript and protein levels, which is observed in other contexts (Maier et

al, 2009), is probably due to multiple factors which involve the transcriptional and translational regulation steps and/or the differential half-lives of proteins and mRNAs under MG132 treatment.

Figures 4A and F - what constitutes a + vs - scoring? What is the threshold? How many times did you perform this experiment?

The – sign corresponding to the molecule indicated to the left on the same line indicates that this molecule was not used to treat the cells and that only its solvent was added (exclusively DMSO as a negative control). On the contrary, the + sign indicates that the molecule indicated to the left on the same line was used to treat the cells.

The threshold to which are compared the various conditions corresponds to the condition where the cells have been treated with a volume of DMSO equal to that of the active molecule used for the treatment. This threshold has been set at 100% under the control condition.

To clarify figures understanding, the captions were modified in the revised version of the manuscript.

Figure 4A and F - in the graph, what are the X-axis points? Time points, different treatments?

X-axis Corresponds to the quantitative measurement of protein expression levels (through the image J software) relative to control conditions (indicated with – sign and treated with DMSO, well N°1) in the different treatment conditions observed in the Western-blot above. Well numbers have been added above the figures and below quantifications to better clarify the understanding. The legend of Figure 4A and 4G (formerly 4F) was also modified to bring more detail.

Figure 5 - there are automated methods for scoring nuclei now. You used an older method of "eyeballing" the nuclei. Can you automate this?

We thank the referee for this suggestion. The quantification of abnormal nuclear morphology was carried out by two methods: manual counting and automated counting (using the Nuclear Irregularity Index (NII) plugin of the Image J Software (Version 1.6.0, NIH, USA)) giving the same results. We have, as suggested, modified this section in the material and methods section and in the legend of Fig. 5B.

Figure 5D - did you show that the untreated or DMSO-treated cultures increased senescence over the 10 days? I don't see this in the graph. Yet you state in the text that you "rescued" senescence. This does not support a rescue of senescence but perhaps a decrease with treatment in the number of senescing cells. In fact, day 10 culture has about the same amount of senescence in the treated culture as day 1. Please respond.

We thank the referee for these remarks. We have considered this comment, and we carried out statistical tests to compare at each time point the evolution of senescence rate relative to the days 0. Indeed, the results show that untreated or DMSO-treated cultures increased senescence over the 10 days. We agree that the results presented in Fig.5D does not support a rescue of senescence but a decrease with treatment in the number of senescing cells.

We modified the figure as well as our statement in the results section accordingly.

Figure 5F - I assume the * indicates comparison with DMSO bar. However, that bar starts out lower on day 1 than MG132. Therefore, I'm concerned that you are not accurate when you state that the relative difference on day 10 indicates improved viability with MG132. The same holds for 5E. You need to show that these cultures start equally, pre-treatment and you have not done that.

We agree that the data could, in this form, be confusing. We improved the graphs by adding cell proliferation and viability rates before treatment (Day 0) to show that these cultures start equally. We also agree that the results presented in the old form does not support an improved proliferation or viability over time but an increase with treatment in the number of viable and proliferating cells on day 10 relative to DMSO-treated cells at the same time point.

We modified the figure as well as our statement in the results section accordingly. In addition, using a conventional cell proliferation assay (BrdU), we show that MG132 treatment scheme increased proliferation rate of HGPS cells with a curve approximately similar, but without reaching the proliferation rate of age-matched WT control cells (new Fig 5E).

Fig 6A - again you need to state or show staining in control line to prove progerin specificity of your antibody.

We thank the referee for expressing this concern. As requested, we provide new image (Fig. 6A) showing negative progerin staining on iPS-VSMC derived from cells line of a healthy individual.

Figure 7 - please make the Y axes for B and C similar. It is hard to compare these two doses otherwise. Why are your untreated bars so different from B to C? Shouldn't they be similar, as they come from untreated muscle in mice with the same mutation? Please address this important issue.

As requested, we made the Y axes for B and C similar. The differences between the two groups of untreated mice are due to inter-individual variability, in fact we observed a high variability in progerin expression levels at baseline even between littermate HGPS mice. These intriguing observations are in agreement with previous publications showing phenotypic, genetic and epigenetic variation among inbred mouse littermates (Gartner, 1990; Oey et al, 2015; Wong et al, 2005). Since epigenetic regulation not only determines what parts of the genome are expressed, but also how they are spliced (Luco et al, 2011) and given that progerin is generated by splicing mechanism, one would expect that progerin amounts vary from one group to another. In this context, the local approach we chose to investigate MG132 effect *in vivo* freed our observation from the variable baseline progerin levels observed among mice.

As noted, lamin A is not affected significantly, though the prior description of splicing effect would lead us to hypothesize that lamin A splicing would increase.

As we have answered above, the reduction of lamin A transcripts is not surprising given that the lamin A 5'SS is also recognized by SRSF-1 independently of the HGPS mutation that promotes the reinforcement of another weak 5' splice-site also recognized by SRSF-1 and leading to progerin production. SRSF1 elimination has already been described to decrease progerin, lamin A and lamin C transcripts levels in HGPS cells upon treatment with Metformin (Egesipe et al, 2016). Consistent with the increase in lamin A/C protein levels that we obtained *in vitro* and which are not significant *in vivo*, the poor correlation between transcript and protein levels, which is observed in other contexts (Maier et al, 2009), is probably due to multiple factors which involve the transcriptional and translational regulation steps and/or the differential half-lives of proteins and mRNAs under MG132 treatment.

Progerin is suppressed much more in C at the higher dose, but SRSF-1 is not. SRSF-1 suppression is similar between B and C. I would expect an increased SRSF-1 suppression to move along with increased progerin splicing suppression. This argues against the influence of SRSF-1.

We thank the referee for these remarks. We agree that we would expect an increased SRSF-1 suppression to move along with increased progerin splicing suppression. However, since MG132 influences other pathways involved in progerin elimination (autophagy, SRSF-5, caspases), it is conceivable that these pathways are not affected in the same way by increasing MG132 concentration. In addition, as answered above, genetic, epigenetic and splicing events variation among mice do not allow to expect linearity between SRSF-1 and progerin levels.

Discussion:

Once you address the issues with data and data presentation, please review the discussion points to assure they accurately reflect the results. Most particularly, your discussion points about the SRSF-1 splicing effect, but also the other issues raised above. You should clarify your data on shape, senescence, viability and proliferation, before stating that these significantly improved with MG132 treatment, or soften that statement in the discussion.

We thank the referee for bringing up these relevant remarks. These points are now highlighted in the discussion of the revised manuscript.

You state that progerin cannot be degraded when it is sumoylated. I think this is an overstatement. Please soften to state that it is not degraded by similar means when it is sumoylated vs when it is not.

We thank the referee for these relevant suggestions. As requested, we replaced our statement on this issue.

To better determine whether progerin sumoylation is required for its clearance, we treated HGPS fibroblasts with sumoylation inhibitors (either 2D08 or ginkgolic acid) in combination with proteasome inhibitors (MG132, MG262, MG115, Bortezomib or Carfilzomib). The results presented in new Fig EV5E show that progerin levels are not significantly affected in the presence or absence of sumoylation inhibitors in response to treatment with MG132, MG115 or Bortezomib. Under the same conditions, the effect of MG262 on progerin clearance is more pronounced when sumoylation is inhibited. On the contrary, sumoylation of progerin appears to be required to achieve its clearance in response to Carfilzomib. Altogether, these results indicate that progerin is not degraded by similar means when it is sumoylated vs when it is not.

You state the MG132 can have benefit without toxicity in vivo. Do you refer only to the references in the paragraph to make this statement? Please clarify the statement.

This refers to several references showing a beneficial effect of MG132 without toxic effects. It should also be mentioned that, when mice were injected with MG132, no deleterious effects were observed (no necrosis or inflammation, nor tumor formation at the injection sites and no weight loss or premature death upon local treatment). But anyway, systemic injections of MG132 in *Lmna*^{G609G/G609G} mice, which of course will be needed to target the different organs involved in the pathophysiology of progeria, will likely require the setting up of an appropriate galenic form of this short peptide which we should test efficacy, toxicity and side effects.

Mice did not show any necrosis or tumor formation at injection sites nor weight loss nor premature death upon local treatment.

The paragraph on IM MG132 treatment does not address the issues raised above regarding results. Please do so.

We apologize for this omission. we have addressed these issues in the revised manuscript.

Also, I don't see how lamin A "compensates" for lamin C reduction. This global statement should be omitted.

As requested, this statement was omitted.

References:

There is a more recent and larger study of mean age of death in progeria which calculates 14.6 years (Gordon et al, 2012). Please update this information appropriately.

We thank the referee for pointing this out. As suggested we updated this information in the revised manuscript.

Primary fibroblasts from HGPS patients exhibit reduced proliferation as well as premature senescence (Huang et al, 2005). This paper does not show these properties in relation to normal fibroblasts. In fact, only HGPS cells are used in this paper. Please improve your reference.

We thank the referee for noting this error. It has been corrected by adding the following references (Goldman et al, 2004; Scaffidi & Misteli, 2005; Scaffidi & Misteli, 2008; Vidak & Foisner, 2016).

I didn't perform an exhaustive reference check, so I urge the authors to please do so since I did find two mistakes fairly easily.

As requested, the references section has been verified and additional references were made in the revised manuscript.

Referee #3 (Remarks):

The manuscript by Harhour and co-workers shows that the proteasome inhibitor MG132 - and some related compounds - surprisingly reduces the levels of progerin in cells from HGPS patients. The proposed mechanism is that MG132 induces autophagy which leads to progerin degradation and that it simultaneously reduces progerin production by interfering with SRSF-1 - a protein that stimulates progerin splicing from the prelamin A message. MG132 thereby improved some of the cellular phenotypes of HGPS, such as the impaired proliferation and reduced viability. Overall, the experiments are well designed and the results provide new insight into the biology of progerin handling in cells and indirectly into the pathogenesis of HGPS. This paper continues a long line of novel discoveries from this leading group of investigators in the field. There are some concerns. One concern - which is difficult to address at this point -- is that the paper contains several different findings, several of which are left unexplained and others which are not coupled to the main story. Therefore, the paper appears a bit unfocused. Other than that, there are a few points that can be addressed:

We thank this referee for his/her comments with constructive suggestions and now address all points in the revised manuscript.

Major:

1. A puzzling result in this paper is that many of the figures (e.g., 1A, C, and D; 4A and F; 6A and B etc.) demonstrate that MG132 dramatically reduces progerin levels; in a substantial proportion of western blots, progerin is essentially gone. Regardless, the effect of MG132 treatment on all of the well-known cellular phenotypes of HGPS cells is modest (e.g., Fig 5). Reducing the levels of a toxic lamin A mutant (progerin in some studies, prelamin A in others) produces much more dramatic effects in earlier studies. So, why didn't the substantial reduction in progerin by MG132 result in more robust cellular effects? Could beneficial progerin-reducing MG132 effects on HGPS cells be masked by other deleterious drug-effects? This needs to be addressed with both experiments and discussion. For example, compare the effect of MG132 on cell phenotypes with cells where progerin expression is reduced with si- or sh-RNAs targeting progerin/prelamin A? (to the same levels as in MG132-treated cells). And, would MG132 treatment reduce the levels of full-length prelamin A that accumulate in patients with related progeroid disorders (where splicing is not involved)?

We thank this referee for this justified criticism and provide additional experiments on this issue. Using a conventional cell proliferation assay (BrdU), we show that MG132 treatment scheme increased proliferation rate of HGPS cells with a curve approximately similar, but without reaching the proliferation rate of age-matched WT control cells (New Fig 5E). In addition, we investigated the rescue of additional HGPS cellular phenotypes (γ -H2AX, Tri-Me-K9, LAP2 α , lamin B1 and HPI α). As presented in the new Fig.5H, treatment of HGPS cells with MG132 improved these cellular phenotypes with a rescue level comparable to that of age-matched WT control cells or HGPS cells where progerin expression is reduced with antisense oligonucleotides (AON) targeting progerin. Furthermore, we performed an RNA-seq experiment and analyzed the expression levels of genes that are misregulated in HGPS fibroblasts. Interestingly, we found an increase in the transcripts levels of genes involved in autophagy activation (p62, ATG4A, ATG4B, ATG4D, ATG3, ATG14, LLK1), proteasome subunits production (PSMC4, PSMB3), splicing (SRSF5), metalloprotease transcripts including MMP3 (whose levels are decreased in HGPS cells), growth factors (EGF, FGF17, FGF22), Apoptosis inhibitors (BAG3, BFAR) as well as the improvement of other transcripts involved in inflammation (IkB, SIRT6) (New Fig 5I).

Regarding the effect of MG132 on prelamin A levels in patients with related progeroid disorders, we have shown that MG132 treatment resulted in a reduction of prelamin A levels, both at the transcriptional and protein levels, in fibroblasts from MAD-B patient (mandibuloacral dysplasia) carrying a homozygous FACE1 c.1274T>C mutation. MG132 treatment also induced a decrease of prelamin A Δ 35, Δ 50 (progerin) and Δ 90 levels in HGPS-like fibroblasts carrying mutations near the donor splice site of exon 11 of the *LMNA* gene (heterozygous c.1968+2T>C, c.1968+1G>A, c.1968+5G>C, c.1968G>A, c.1868C>G) (Barthelemy et al, 2015; Harhour et al, 2016). These data will be shown in another article.

2. In Fig. 1D; did the chloroquine treatment result in significantly higher levels of progerin compared to MG132 alone or do the stars only concern the comparison with the high levels or progerin in untreated HGPS cells?

We thank the referee for this important remark and apologize for this omission. Progerin levels in the cells treated either with MG132 alone or with MG132 + Chloroquine are compared to the conditions where the cells are treated with DMSO (the significance of the statistical tests is indicated by "*"). We have added statistical tests for the comparison of progerin levels under conditions where the cells are treated with MG132 + Chloroquine compared to the condition where the cells are treated with MG132 for 48 h (the statistical tests are indicated by "\$"). We modified the figure and its caption accordingly.

3. Lines 352-356: The mechanisms underlying MG132 effects on progerin levels needs to be addressed and discussed better because the 5.5-page discussion doesn't do the job of succinctly describing it. You should use the Fig EV7 more effectively in combination with the text to accomplish this.

If the effect is unrelated to proteasome inhibition, by what mechanisms would MG132, but not BTZ or CFZ, stimulate macroautophagy or nuclear-to-cytoplasm shuttling? Because the chloroquine effect was modest, is the MG132-effect mainly mediated by the reduced SRSF-1 levels and the reduced progerin mRNA levels?

We thank this referee for bringing up this important question. MG132, Bortezomib and Carfilzomib differ structurally, mechanistically and physicochemically (Bortezomib: Boronate class. Carfilzomib: Epoxyketones class and MG132: aldehyde class), they also differ in their inhibition on the 3 proteasome activities (Fig. EV4D) and in their action reversibility (Britton et al, 2009; Kisselev et al, 2012). This indicates that these 3 molecules could have different effects. In addition, it has been shown that Bortezomib inhibits autophagy (Kao et al, 2014; Periyasamy-Thandavan et al, 2010), hence probably its inability to induce an effective progerin clearance compared to that of MG132. On the other hand, we did not observe a decrease of SRSF-1 in the presence of Bortezomib or Carfilzomib, in contrast to MG132 (data not shown). In addition, we have also shown that these molecules do not respond in the same way to sumoylation inhibitors (New Fig. EV5E). All these observations are in agreement with a possible action variability of these three molecules. The effect of MG132 resulted from a set of actions (activation of autophagy, reduction of SRSF-1 and increase of SRSF-5) as evidenced by the complete restoration of progerin levels in the presence of autophagy inhibitors and of caspases (inhibiting SRSF-1 cleavage). These three actions are not provided by Bortezomib or Carfilzomib. We would like to mention that the effect of chloroquine is not modest since it restores an average of 40% of progerin levels (Fig. 1D: MG132+Chloroquine 50µM vs 48H MG132; Fig. 4A: Well 2 vs Well 4; Fig. 4G: Well 2 vs Well 6).

4. Does the MG132 effect require progerin to be present in abnormal PMLs? Would MG132 treatment reduce progerin levels in early passage cells, before the appearance of abnormal PMLs?

We thank the referee for pointing this out. We confirm that the MG132 effect does not require progerin to be present in abnormal PML-NBs. Indeed, some experiments were performed at early passages, when the abnormal PML-NBs are not yet present, giving the same results.

5. Data in Fig. 5E and F: With the normalization at each time point, it is difficult to judge the potential cumulative effect of the treatment over time. Conventional cell proliferation assays should be shown here. This is important as you conclude that MG132 significantly enhanced proliferation and reduced senescence of HGPS cell lines.

We agree that the data could, in this form, be confusing. We improved these data by changing the graph type and by adding cell proliferation and viability rates before treatment (Day 0) to show that these cultures start equally. As requested, using a conventional cell proliferation assay (BrdU), we show that MG132 treatment scheme increased proliferation rate of HGPS cells with a curve approximately similar, but without reaching the proliferation rate of age-matched WT control cells (New Fig 5E). We also investigated the rescue of additional HGPS cellular phenotypes (γ -H2AX, Tri-Me-K9, LAP2 α , lamin B1 and HP1 α). As presented in the new Fig.5H, treatment of HGPS cells with MG132 improved these cellular phenotypes with a rescue level comparable to that of age-matched WT control cells or HGPS cells where progerin expression is reduced with antisense oligonucleotides (AON) targeting progerin. Furthermore, we performed an RNA-seq experiment and analyzed the expression levels of genes that are misregulated in HGPS fibroblasts. Interestingly, we found an increase in the transcripts levels of genes involved in autophagy activation (p62,

ATG4A, ATG4B, ATG4D, ATG3, ATG14, LLK1), proteasome subunits production (PSMC4, PSMB3), splicing (SRSF5), metalloprotease transcripts including MMP3 (whose levels are decreased in HGPS cells), growth factors (EGF, FGF17, FGF22), Apoptosis inhibitors (BAG3, BFAR) as well as the improvement of other transcripts involved in inflammation (IkB, SIRT6) (New Fig 5I).

6. Lines 7, 93, and 335: A biomarker that does not appear in the cells until after 12-20 passages is not particularly useful. The late appearance of the abnormal PML-NBs should be commented on when mentioning biomarkers. Their late appearance in cell-culture dishes also suggest the possibility that they are not relevant *in vivo*. However, they are clearly specific to the HGPS cells and the careful analyses of their contents and dynamics allowed the investigators to uncover important new biology.

We thank this referee for this justified criticism. We agree that such abnormal PML-NBs, which are nonetheless specific to the HGPS cells, could be useless *in vivo*, since they appear at late passages in culture. Our statement has been modified accordingly by adding "in late passage HGPS cell lines".

7. Many of the nuclei shown in regular and expanded view figures show only minor shape abnormalities. Can you comment on whether the findings on progerin localization/co-localization hold up in grossly abnormally-shaped nuclei and is there a correlation between abnormal PMLs and blebbing?

We thank the referee for pointing this out. We confirm that progerin localization / co-localization experiments were performed on cells at different passages including very late passages where the increase in nuclear blebbing is correlated with the appearance of abnormal PML-NBs in which progerin remains present.

Minor comments:

1. Fix lots of spelling mistakes: e.g., line 42 "heterodimerize"; line 954 "de"; line 344 "benn"; EVF 1E, "Transpatent"

Thank you for noting these spelling mistakes. They have been corrected.

2. The legend to Figure EV5 B and C do not match the labeling in the figure (they have been switched).

Thank you for noting these errors. They have been corrected.

3. Lines 954, 955: was the medium replaced with new drug every 48 hours? If so, change the text to make that clear.

Thank you for mentioning this point. Indeed, the medium was replaced with new drug every 48 hours. This is now specified in the revised version of the manuscript.

References

Barthelemy F, Navarro C, Fayek R, Da Silva N, Roll P, Sigaudy S, Oshima J, Bonne G, Papadopoulou-Legbelou K, Evangeliou AE et al (2015) Truncated prelamin A expression in HGPS-like patients: a transcriptional study. *Eur J Hum Genet* 23: 1051-1061

Boban M, Foisner R (2016) Degradation-mediated protein quality control at the inner nuclear membrane. *Nucleus-Phila* 7: 41-49

Britton M, Lucas MM, Downey SL, Screen M, Pletnev AA, Verdoes M, Tokhunts RA, Amir O, Goddard AL, Pelphrey PM et al (2009) Selective inhibitor of proteasome's caspase-like sites sensitizes cells to specific inhibition of chymotrypsin-like sites. *Chem Biol* 16: 1278-1289

Butler JT, Hall LL, Smith KP, Lawrence JB (2009) Changing Nuclear Landscape and Unique PML Structures During Early Epigenetic Transitions of Human Embryonic Stem Cells. *J Cell Biochem* 107: 609-621

Dou Z, Xu C, Donahue G, Shimi T, Pan JA, Zhu J, Ivanov A, Capell BC, Drake AM, Shah PP et al (2015) Autophagy mediates degradation of nuclear lamina. *Nature* 527: 105-109

Doucas V, Evans RM (1996) The PML nuclear compartment and cancer. *Bba-Rev Cancer* 1288: M25-M29

Fukuda M, Asano S, Nakamura T, Adachi M, Yoshida M, Yanagida M, Nishida E (1997) CRM1 is responsible for intracellular transport mediated by the nuclear export signal. *Nature* 390: 308-311

Gartner K (1990) A 3rd Component Causing Random Variability Beside Environment and Genotype - a Reason for the Limited Success of a 30 Year Long Effort to Standardize Laboratory-Animals. *Lab Anim* 24: 71-77

Goldman RD, Shumaker DK, Erdos MR, Eriksson M, Goldman AE, Gordon LB, Gruenbaum Y, Khuon S, Mendez M, Varga R et al (2004) Accumulation of mutant lamin A causes progressive changes in nuclear architecture in Hutchinson-Gilford progeria syndrome. *P Natl Acad Sci USA* 101: 8963-8968

Harhour K, Navarro C, Baquerre C, Da Silva N, Bartoli C, Casey F, Mawuse GK, Doubaj Y, Levy N, De Sandre-Giovannoli A (2016) Antisense-Based Progerin Downregulation in HGPS-Like Patients' Cells. *Cells* 5

Jung YS, Lee SJ, Lee SH, Chung JY, Jung YJ, Hwang SH, Ha NC, Park BJ (2013) Loss of VHL promotes progerin expression, leading to impaired p14/ARF function and suppression of p53 activity. *Cell Cycle* 12: 2277-2290

Kao C, Chao A, Tsai CL, Chuang WC, Huang WP, Chen GC, Lin CY, Wang TH, Wang HS, Lai CH (2014) Bortezomib enhances cancer cell death by blocking the autophagic flux through stimulating ERK phosphorylation. *Cell Death Dis* 5: e1510

King MC, Lusk CP (2016) A model for coordinating nuclear mechanics and membrane remodeling to support nuclear integrity. *Curr Opin Cell Biol* 41: 9-17

Kisselev AF, van der Linden WA, Overkleeft HS (2012) Proteasome inhibitors: an expanding army attacking a unique target. *Chem Biol* 19: 99-115

Korolchuk VI, Menzies FM, Rubinsztein DC (2010) Mechanisms of cross-talk between the ubiquitin-proteasome and autophagy-lysosome systems. *FEBS letters* 584: 1393-1398

Lee SJ, Jung YS, Yoon MH, Kang SM, Oh AY, Lee JH, Jun SY, Woo TG, Chun HY, Kim SK et al (2016) Interruption of progerin-lamin A/C binding ameliorates Hutchinson-Gilford progeria syndrome phenotype. *J Clin Invest* 126: 3879-3893

Luco RF, Allo M, Schor IE, Kornblihtt AR, Misteli T (2011) Epigenetics in Alternative Pre-mRNA Splicing. *Cell* 144: 16-26

Luo M, Zhao X, Song Y, Cheng H, Zhou R (2016) Nuclear autophagy: An evolutionarily conserved mechanism of nuclear degradation in the cytoplasm. *Autophagy* 12: 1973-1983

Maier T, Guell M, Serrano L (2009) Correlation of mRNA and protein in complex biological samples. *FEBS letters* 583: 3966-3973

Oey H, Isbel L, Hickey P, Ebaid B, Whitelaw E (2015) Genetic and epigenetic variation among inbred mouse littermates: identification of inter-individual differentially methylated regions. *Epigenet Chromatin* 8

Periyasamy-Thandavan S, Jackson WH, Samaddar JS, Erickson B, Barrett JR, Raney L, Gopal E, Ganapathy V, Hill WD, Bhalla KN et al (2010) Bortezomib blocks the catabolic process of autophagy via a cathepsin-dependent mechanism, affects endoplasmic reticulum stress and induces

caspase-dependent cell death in antiestrogen-sensitive and resistant ER+ breast cancer cells. *Autophagy* 6: 19-35

Scaffidi P, Misteli T (2005) (R)eversal of the cellular phenotype in the premature aging disease Hutchinson-Gilford progeria syndrome. *Nat Med* 11: 440-445

Scaffidi P, Misteli T (2008) Lamin A-dependent misregulation of adult stem cells associated with accelerated ageing. *Nat Cell Biol* 10: 452-U167

Slee EA, Adrain C, Martin SJ (2001) Executioner caspase-3, -6, and -7 perform distinct, non-redundant roles during the demolition phase of apoptosis. *J Biol Chem* 276: 7320-7326

Sternsdorf T, Grotzinger T, Jensen K, Will H (1997) Nuclear dots: Actors on many stages. *Immunobiology* 198: 307-331

Stixova L, Matula P, Kozubek S, Gombitova A, Cmarko D, Raska I, Bartova E (2012) Trajectories and nuclear arrangement of PML bodies are influenced by A-type lamin deficiency. *Biol Cell* 104: 418-432

Tang B, Cai J, Sun L, Li Y, Qu J, Snider BJ, Wu S (2014) Proteasome inhibitors activate autophagy involving inhibition of PI3K-Akt-mTOR pathway as an anti-oxidation defense in human RPE cells. *PLoS one* 9: e103364

Vautrot V, Aigueperse C, Oillo-Blanloeil F, Hupont S, Stevenin J, Branlant C, Behm-Ansmant I (2016) Enhanced SRSF5 Protein Expression Reinforces Lamin A mRNA Production in HeLa Cells and Fibroblasts of Progeria Patients. *Human mutation* 37: 280-291

Vidak S, Foisner R (2016) Molecular insights into the premature aging disease progeria. *Histochem Cell Biol* 145: 401-417

Wong AHC, Gottesman II, Petronis A (2005) Phenotypic differences in genetically identical organisms: the epigenetic perspective. *Hum Mol Genet* 14: R11-R18

Zang Y, Thomas SM, Chan ET, Kirk CJ, Freilino ML, DeLancey HM, Grandis JR, Li C, Johnson DE (2012) The next generation proteasome inhibitors carfilzomib and oprozomib activate pro-survival autophagy via induction of the unfolded protein response and ATF4. *Autophagy* 8: 1873-1874

Zhu K, Dunner K, Jr., McConkey DJ (2010) Proteasome inhibitors activate autophagy as a cytoprotective response in human prostate cancer cells. *Oncogene* 29: 451-462

2nd Editorial Decision

19 May 2017

Thank you for the submission of your revised manuscript to EMBO Molecular Medicine. We have now received the enclosed reports from the referees that were asked to re-assess it. As you will see the reviewers are now globally supportive and I am pleased to inform you that we will be able to accept your manuscript pending the following final amendments:

1) Please address the remaining issues commented by referee 1. We already discussed the point 1 [supporting your decision to leave the data in], but regarding point 2, we would like to encourage you to address in writing the concerns of this referee, maybe tuning down the claims and quantifying the rescue phenotype from more than 1 image as suggested would be ideal. Please provide a letter INCLUDING the reviewer's reports and your detailed responses to their comments (as Word file).

2) We note that you currently have 7 main figures and 7 EV figures, which are 2 too many. Please try to merge some of the EV figures, or prepare an Appendix file and move your extra supplementary figures there. Please see our guidelines for information on how to format the Appendix file and make sure to update the call outs:

<http://embomolmed.embopress.org/authorguide#expandedview>

3) M&M: fibroblasts reprogramming (p26): it is said there that you sometimes used purchased fibroblasts but sometimes used some taken from biopsies of patients hospitalised in Marseille. For those samples we absolutely need the right ethical agreement (please see our guidelines), along with informed consent (see: <http://embomolmed.embopress.org/authorguide#humansubjects>)

4) Please deposit the RNAseq data to the appropriate repositories and provide an accession number to be included within the article and Author's checklist. This is mandatory for publication.

Please submit your revised manuscript within two weeks. I look forward to seeing a revised form of your manuscript as soon as possible.

***** Reviewer's comments *****

Referee #1 (Comments on Novelty/Model System):

Proposed mechanism is not supported by the presented data.

Referee #1 (Remarks):

In the revised manuscript EMM-2016-07315-V2, the authors only minimally address the issues raised in the original review. Very little additional data is included and the authors mostly attempt to argue the points of concern away. The authors have not sufficiently addressed the points raised in the original review and consequently we can still not recommend this manuscript for publication.

Two major points were made in our original review:

1) the manuscript was not well structured and contained seemingly unrelated pieces of data. The manuscript still contains large amounts of disparate data and combines MG132 and PML-NBs studies with little clear connection. The authors' justification that they want to present a complete story does not justify this collection of only loosely related data.

2) the proposed mechanism is still not convincingly demonstrated. In response to our inquiry as to how SRSF1 mediates the observed effects, the authors now introduce a second splicing factor SRSF5 as an effector. However, it similarly does not explain the observed effect, largely for the same reasons cited in our original comments. The proposed mechanism is highly speculative. In addition, the authors did not include any additional studies to probe how MG132 promotes autophagy except providing theoretical explanations. No additional experimental data is provided. Furthermore, the data showing the rescue of HGPS cellular defects (LAP2, LB1, HP1 α , Tri-Me-K9, γ -H2AX) are insufficient as they only show a single IF image for each phenotype authors rather than quantitative data.

Of concern is also the fact that the authors still emphasize MG132 as a promising potential therapy for HGPS, despite the lack of mechanistic insight into its action or in vivo studies. This is a premature statement.

Overall, this manuscript is not suitable for publication due to the absence of clear mechanistic insight and unsubstantiated claims regarding clinical relevance of the reported observations and interrogated compounds.

Referee #2 (Remarks):

The authors have addressed my concerns adequately. The paper will stimulate interest, and follow-up studies for the field of HGPS.

One minor issue - line 320 Apoptosis should have small a.

Referee #3 (Remarks):

The authors have answered my concerns.

2nd Revision - authors' response

02 June 2017

Referee #1 (Comments on Novelty/Model System):

Proposed mechanism is not supported by the presented data.

Referee #1 (Remarks):

In the revised manuscript EMM-2016-07315-V2, the authors only minimally address the issues raised in the original review. Very little additional data is included and the authors mostly attempt to argue the points of concern away. The authors have not sufficiently addressed the points raised in the original review and consequently we can still not recommend this manuscript for publication.

Two major points were made in our original review:

1) the manuscript was not well structured and contained seemingly unrelated pieces of data. The manuscript still contains large amounts of disparate data and combines MG132 and PML-NBs studies with little clear connection. The authors' justification that they want to present a complete story does not justify this collection of only loosely related data.

2) the proposed mechanism is still not convincingly demonstrated. In response to our inquiry as to how SRSF1 mediates the observed effects, the authors now introduce a second splicing factor SRSF5 as an effector. However, it similarly does not explain the observed effect, largely for the same reasons cited in our original comments. The proposed mechanism is highly speculative. In addition, the authors did not include any additional studies to probe how MG132 promotes autophagy except providing theoretical explanations. No additional experimental data is provided. Furthermore, the data showing the rescue of HGPS cellular defects (LAP2, LB1, HP1±, Tri-Me-K9, Î⁻-H2AX) are insufficient as they only show a single IF image for each phenotype authors rather than quantitative data.

Of concern is also the fact that the authors still emphasize MG132 as a promising potential therapy for HGPS, despite the lack of mechanistic insight into its action or in vivo studies. This is a premature statement.

Overall, this manuscript is not suitable for publication due to the absence of clear mechanistic insight and unsubstantiated claims regarding clinical relevance of the reported observations and interrogated compounds.

We would like to mention that, after our discussion and your agreement to maintain the exploration of the PML-NBs and the interest that the 2nd referee has on this part of our story, as agreed, we have kept it in the revised version of the manuscript.

Contrary to what was stated in referee # 1 remarks, we think that we have largely addressed each point raised in the initial review by adding novel experiments and explanations. In particular, the answer to the question "**How SRSF1 mediates the observed effects**" is trivial since this data has been already described in the literature (Lopez-Mejia et al., 2011). In addition, we have shown that:

1. SRSF-1 levels decrease are concomitant with progerin levels decrease (Fig. 4C).
2. SRSF-1 Silencing by siRNA experiments induces a decrease in progerin levels (Fig. 4F)
3. Inhibition of SRSF-1 cleavage in the presence of PAN caspase inhibitors allows restoration of SRSF-1 levels.

Concerning the comment "**Given a half-life of more than 24 hours for progerin, these are clearly not SRSF1-mediated effect**", we also provided a clear answer indicating that the half-life of progerin does not prevent progerin turnover due to neosynthesis linked to splicing regulation, and degradation. Furthermore, it has been shown that the half-life of progerin described in Youn-Sang Jung paper (approximately 20h) is largely decreased to 3.5 hours following treatment with the von Hippel-Lindau tumor suppressor also known as pVHL (Jung et al, 2013), or by a compound that

blocks progerin–lamin A/C binding (Lee et al, 2016). These observations are consistent with a possible change in progerin half-life in response to MG132 treatment. Moreover, as shown in Fig. 1C, treatment with DMSO does not induce any decrease in progerin levels under control conditions. Thus, eliminating a possible involvement of progerin turnover in its clearance.

Moreover, we added further data proving that MG132 acts not only at the degradation step, by activating autophagy, but also at progerin synthesis by modulating SRSF-1 and SRSF-5. In particular, we showed that splicing modulation by these two splicing cofactors is complementary and sequential, with SRSF-5 increase and SRSF1 decrease both downregulating progerin synthesis. Overall, It is thus clear that MG132 treatment does modify progerin half-life.

In response to the comment **“The authors did not include any additional studies to probe how MG132 promotes autophagy”** we would like to remember that autophagy activation by MG132 has been widely described in the literature. Indeed, the proteasome inhibitors induce autophagy as a compensatory response (Tang et al, 2014; Zang et al, 2012; Zhu et al, 2010). Several potential explanations have been suggested to explain the exact mechanism(s) of this cross-talk. One such proposed mechanism involving induction of the unfolded protein response (UPR) or inhibition of mTOR and induction of damage-regulated autophagy modifier (Korolchuk et al, 2010). Further, we showed in our manuscript that autophagy activation is supported by the increased amounts of LC3B-II/LC3BI ratios on immunoblotting assays, progerin delocalization into cytoplasmic autophagic vacuoles, increased autophagic transcript levels using additional RNA seq experiments and partial restoration of progerin levels in presence of Chloroquine or Bafilomycin A1 which are commonly used as autophagy inhibitors.

"The data showing the rescue of HGPS cellular defects (LAP2, LB1, HP1 +, Tri-Me-K9, H2-AX) are insufficient as they only show a single IF image for each phenotype authors rather than quantitative data". As suggested, we quantified the rescue phenotype from more than 1 image. We modified the figure 5H accordingly.

Referee #2 (Remarks):

The authors have addressed my concerns adequately. The paper will stimulate interest, and follow-up studies for the field of HGPS.

One minor issue - line 320 Apoptosis should have small a.

Thank you for your supportive report
The spelling mistake has been corrected.

Referee #3 (Remarks):

The authors have answered my concerns.

Thank you for your supportive report

Corresponding Author Name: Nicolas Levy

Manuscript Number: Manuscript EMM-2016-07315